# Ozone as an environmental driver of influenza

Fang Guo [1,11], Pei Zhang[1,11], Vivian Do [2,11], Jakob Runge[3,4], Kun Zhang[5,6], Zheshen Han[1], Shenxi Deng [1], Hongli Lin[1], Sheikh Taslim Ali [1,7], Ruchong Chen[8], Yuming Guo [9] & Linwei Tian [1,10] ✉

Under long-standing threat of seasonal influenza outbreaks, it remains imperative to understand the drivers of influenza dynamics which can guide mitigation measures. While the role of absolute humidity and temperature is extensively studied, the possibility of ambient ozone ($O_3$) as an environmental driver of influenza has received scant attention. Here, using state-level data in the USA during 2010–2015, we examined such research hypothesis. For rigorous causal inference by evidence triangulation, we applied 3 distinct methods for data analysis: Convergent Cross Mapping from state-space reconstruction theory, Peter-Clark-momentary-conditional-independence plus as graphical modeling algorithms, and regression-based Generalised Linear Model. The negative impact of ambient $O_3$ on influenza activity at 1-week lag is consistently demonstrated by those 3 methods. With $O_3$ commonly known as air pollutant, the novel findings here on the inhibition effect of $O_3$ on influenza activity warrant further investigations to inform environmental management and public health protection.

Influenza imposes a great health and economic burden worldwide, killing about half a million people annually[1,2]. To ease the threat of seasonal and novel influenza epidemics, improved understanding of potential environmental drivers of influenza dynamics has been a research priority. Influenza activity is affected by multi-dimensional determinants, including antigenic drift, host susceptibility[3], social factors (e.g., population-mixing and contact rates)[4], and environmental conditions[5]. Previous epidemiological and experimental studies have examined the relationships of absolute humidity (AH) and temperature (T) with influenza[6,7]. Recently, a negative association between daily ambient ozone ($O_3$) and influenza transmissibility has

also been reported in a time series study of Hong Kong[8]. This human population finding of Hong Kong is consistent with some available laboratory and clinical evidence that the commonly known air pollutant and pulmonary irritant, $O_3$, not only exhibits virucidal potential through its oxidizing power[9,10], but also primes host immunity against viral infection[11–13]. It would be intriguing to see whether the $O_3$-influenza relationship observed in subtropical Hong Kong also holds true in temperate climates. In this work, we used the publicly available weekly state-level data in the USA during 2010–2015 to examine the acute effect of ambient $O_3$ on influenza dynamics— whether a change in weekly ambient $O_3$ leads to a change in influenza

[1]School of Public Health, The University of Hong Kong, Pok Fu Lam, Hong Kong SAR, PR China. [2]Mailman School of Public Health, Columbia University, New York, NY, USA. [3]Deutsches Zentrum für Luft- und Raumfahrt (DLR), Institut für Datenwissenschaften, Jena, Germany. [4]Technische Universität Berlin, Berlin, Germany. [5]Department of Philosophy, Carnegie Mellon University, Pittsburgh, PA, USA. [6]Machine Learning Department, Mohamed bin Zayed University of Artificial Intelligence, Abu Dhabi, UAE. [7]Laboratory of Data Discovery for Health Limited, Hong Kong Science Park, New Territories, Hong Kong SAR, PR China. [8]State Key Laboratory of Respiratory Disease, National Clinical Research Center for Respiratory Disease, National Center for Respiratory Medicine, Guangzhou Institute of Respiratory Health, Department of Allergy and Clinical Immunology, The First Affiliated Hospital of Guangzhou Medical University, Guangzhou, PR China. [9]Climate, Air Quality Research Unit, School of Public Health and Preventive Medicine, Monash University, Melbourne, VIC, Australia. [10]Institute for Climate and Carbon Neutrality, The University of Hong Kong, Pok Fu Lam, Hong Kong SAR, PR China. [11]These authors contributed equally: Fang Guo, Pei Zhang, Vivian Do. ✉e-mail: linweit@hku.hk

activity within 2 weeks in the community when keeping all other variables the same.

The totality of evidence has been proposed as a way forward to strengthen causal inference with observational data[14]: cross-checking multiple methods can enable more robust causal interpretation that is supported by multiple algorithms and conceptual principles but contradicted by none. Here, we propose an integrative methodological framework with three distinct approaches in examining the effect of ambient $O_3$ on influenza dynamics in the USA. Namely, (1) Convergent cross mapping (CCM), a causality test kit based on state-space reconstruction (SSR) for dynamical systems[15,16], (2) a graphical modeling approach called Peter-Clark-momentary-conditional-independence plus (PCMCI+)[17,18], and (3) a statistical regression method Generalized Linear Model (GLM)[19], are used in our study. Since these methods are endowed with disparate theoretical assumptions and hidden biases, we envision that the confidence to make causal inference regarding the scientific question of interest would be strengthened if consistent findings are reached[14,20].

## Results

### Dynamic data of environmental variables and influenza

Weekly time series of environmental variables (namely, $O_3$, AH, and T) and influenza activity ("Flu") in the USA during October 3, 2010 to May 31, 2015 are depicted in Fig. 1. "Flu" is calculated by taking the product of two proportions: the proportion of influenza-like-illness (ILI) cases among all clinical visits in the community and the proportion of influenza-positive specimens, being an arguably good proxy measure of influenza activity in the community (see details in the "Method" section). Seasonality is observed in all the time series, with

influenza activity showing winter peak but summer trough, and environmental variables showing the opposite to certain extent. There are long stretches of 0 values in the influenza time series, especially outside of influenza season, which contain little causal information for exploration. As a consequence, this study only focused on the influenza season (that is considered October through May) in the USA for analysis. The weekly mean level of influenza activity is $4.25 \times 10^{-3}$, which can be understood as 425 expected cases per 100,000 population.

### Environmental drivers of influenza

The estimated effects of environmental factors on influenza activity in the USA using three methods (namely, CCM, PCMCI+, and GLM) are summarised in Table 1. While these methods adopt different measures of effect size, they share one common interpretation rule: a negative value indicates a negative effect size, and vice versa. The negative effect of 1-week lagged AH on influenza activity was detected by PCMCI + and GLM but not CCM, and the negative effect of 2-week lagged air T on influenza was detected by GLM alone. Ambient $O_3$ was found to reduce influenza activity ($p < 1.0 \times 10^{-3}$) at lag 1 (week), consistently by three distinct methods.

CCM under the umbrella of Empirical Dynamic Modeling (EDM) approach was used to conduct the causality test in dynamical systems[15]. The intuition behind CCM is to examine how well a hypothesized driving variable can be cross-mapped (cross-predicted) by the effect variable given putative causal information injected. And such cross-mapping should perform better as more data points (i.e., larger library size) are available to construct the attractor, showing a convergence property, thus the name Convergent Cross-Mapping (CCM)

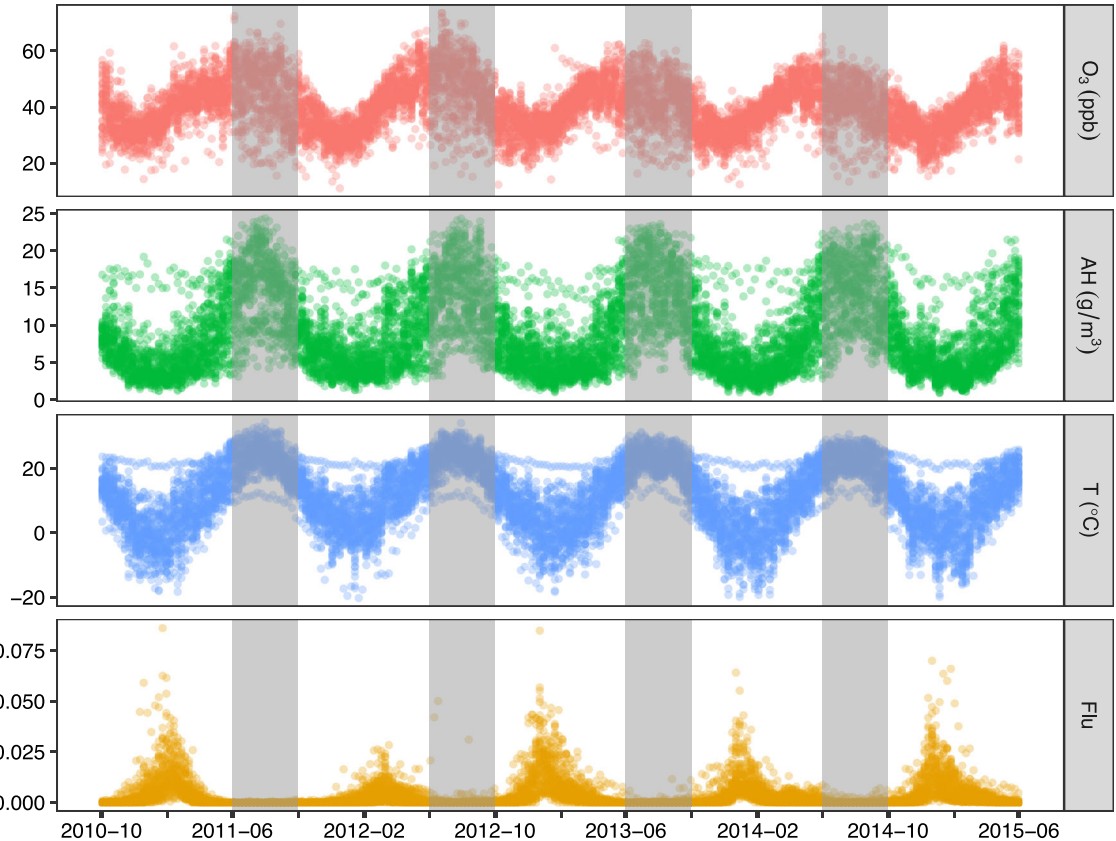

**Fig. 1 | Weekly time series of environmental measurements and influenza activity in 46 states of the USA during 2010–2015.** Influenza activity is indicated by the variable "Flu" which is the product of two proportions: the proportion of influenza-like-illness cases among all clinical consultations in the community and the proportion of laboratory confirmed influenza-positive specimens among all specimens tested. Long stretches of 0 values for "Flu" in non-influenza season (June to September), shaded in gray columns, contain little information for causal inference and thus are omitted from data analysis.

**Table 1 | Effects of environmental factors on influenza activity estimated by three distinct methods, based on weekly state-level data of the USA during 2010–2015**

| | CCM | | PCMCI+ | | GLM | |
|---|---|---|---|---|---|---|
| | **Effect** | **p value** | **Effect** | **p value** | **Effect** | **p value** |
| **Ozone** | | | | | | |
| Lag 0 | **−0.115** | **$1.6 \times 10^{-6}$** | **−0.073** | **$1.2 \times 10^{-9}$** | −0.016 | $5.4 \times 10^{-1}$ |
| Lag 1 | **−0.106** | **$1.9 \times 10^{-7}$** | **−0.057** | **$21 \times 10^{-6}$** | **−0.102** | **$5.9 \times 10^{-5}$** |
| Lag 2 | **−0.112** | **$1.0 \times 10^{-8}$** | −0.030 | $1.2 \times 10^{-2}$ | −0.017 | $3.9 \times 10^{-1}$ |
| **Absolute humidity** | | | | | | |
| Lag 0 | −0.114 | $8.3 \times 10^{-2}$ | 0.010 | $4.0 \times 10^{-1}$ | −0.037 | $4.3 \times 10^{-1}$ |
| Lag 1 | −0.088 | $1.4 \times 10^{-2}$ | **−0.107** | **$4.7 \times 10^{-19}$** | **−0.310** | **$6.7 \times 10^{-8}$** |
| Lag 2 | −0.050 | $2.9 \times 10^{-1}$ | 0.0013 | $9.1 \times 10^{-1}$ | −0.020 | $6.6 \times 10^{-1}$ |
| **Temperature** | | | | | | |
| Lag 0 | −0.132 | $2.6 \times 10^{-1}$ | −0.039 | $1.2 \times 10^{-3}$ | 0.075 | $2.6 \times 10^{-2}$ |
| Lag 1 | −0.113 | $1.4 \times 10^{-1}$ | −0.034 | $4.8 \times 10^{-3}$ | 0.076 | $1.1 \times 10^{-1}$ |
| Lag 2 | −0.080 | $6.4 \times 10^{-1}$ | −0.034 | $4.3 \times 10^{-3}$ | **−0.158** | **$2.0 \times 10^{-6}$** |

In all three sets of results, bold values suggest relationships with statistical significance of $p < 1.0 \times 10^{-3}$. In CCM, causality test against 1000 seasonal surrogates was first performed for each state and those $p$ values were then pooled to obtain meta-significance for nationwide results, while effect was estimated by multivariate S-map analysis. In PCMCI+, state-level data were pooled to obtain one set of results for the nation, and effect is measured by momentary conditional independence (MCI) test with partial correlation method. In GLM, regression with logit link function was first performed for each state, the coefficients of which were then pooled by meta-analysis for nationwide results.

*CCM* convergent cross mapping, *PCMCI+* Peter-Clark-momentary-conditional-independence plus, *GLM* generalized linear model.

for causality test (see details in the "Methods" section). Here, the CCM skill was calculated as the improved prediction accuracy obtained at the maximum library size over that at the minimum library size ($\Delta\rho_{CCM} = \rho_{maxLib} - \rho_{minLib}$). To rule out the influence of shared seasonality between time series, the CCM skill ($\Delta\rho_{CCM}$) of observed time series was tested against surrogate data. Figure 2a presents the state-level CCM surrogate tests on whether there is a causal effect of $O_3$, AH and T at 1-week lag on influenza activity; Fig. 2b presents a summary of measured $\Delta\rho_{CCM}$ of each state and the nationwide meta-significance estimates by summing the logs of state-specific CCM test $p$ values. While the state-level relationships are variable and all three environmental factors could drive influenza activity in some states (as signified by filled red dots), the nationwide test indicates that $O_3$ alone is an environmental driver of influenza activity at the significance threshold of $p_{meta} < 1.0 \times 10^{-3}$; moreover, CCM skill for $O_3$ is significantly greater than that for AH ($p < 2.8 \times 10^{-5}$) and T ($p < 4.3 \times 10^{-5}$) by Wilcoxon rank sum test. When repeating CCM in the nonsensical causal direction by setting the candidate cause and effect in reverse, none of the CCM results is significant (see Supplementary Fig. S1), addressing the concern of spurious relationship generated by noncausal synchrony.

Following CCM causality test, multivariate S-map (sequential locally weighted global linear map) is conducted to quantify the effect magnitude of putative environmental drivers on influenza activity[16,21]. Figure 3 plots the state-specific median effect size of $O_3$, at 1-week lag, on influenza activity onto the map, showing a mostly negative sign across the states. The median effect size is −0.106; that is, one SD increment in $O_3$ (8.3 ppb) leads to a 0.106 SD decrease in logit-transformed influenza activity in the following week. All states see negative effect size except for the two states of Mississippi and Hawaii though they do not pass the CCM causality tests (Fig. 2a). The effect estimates of AH and T on influenza are also negative, but they fail to pass the CCM causality tests (Table 1).

To cement the credibility of making causal inference, a probabilistic graphical modeling framework called PCMCI+ (see detailed description in "Methods" section) is adopted to estimate the causal networks of the underlying system[17,18]. The output of PCMCI+ reads as

a directed acyclic graph (DAG) which can be interpreted as random variables (in nodes) linked by causal dependencies under certain assumptions (Supplementary Table S1). Figure 4a shows the state-by-state dependency networks. Eighteen out of 46 states see a direct link from $O_3$ to influenza activity, of which 17 are of a negative sign and 1 is positive for North Dakota. Thirteen states see direct and/or indirect links from AH to influenza activity: the effect sign is mixed but largely negative. Fifteen states see negative link from T to influenza activity, of which 10 links are direct, but 5 are indirect through $O_3$. Here, the hyperparameter significance level ($\alpha_{PC}$) for iteratively filtering out spurious links was set as 0.05 at the state-level analyses.

Figure 4b shows a summary dependency graph estimated by concatenated time series from individual states; a more stringent criterion $\alpha_{PC}$ of $1.0 \times 10^{-3}$ was used to yield the nationwide graphical output. Ambient $O_3$ bears a direct negative effect on influenza activity at lag 0 (i.e., within the same week) and a lag of 1 week; air T affects influenza activity in a negative manner indirectly through $O_3$ at lag 0. With respect to AH, its total effect on influenza activity is obscure: aside from a direct negative coupling at lag 1, it also has indirect but positive effect through $O_3$. AH and T are strongly coupled with each other, as expected.

Supplementary to the above two causal discovery methods, customary regression method of GLM is conducted at each state to analyze the relationship of environmental factors with influenza activity, adjusting for secular trend, seasonality, as well as inherent autocorrelation. Figure 5a indicates that 1-week lagged statistical associations between each environmental variable and influenza activity are mixed at state level. The state-level regression coefficients are then pooled using a meta-analysis model (Fig. 5b). One SD increment in $O_3$ concentration is associated with a reduction of 0.102 (CI: −0.186, −0.018; $p < 5.9 \times 10^{-5}$) in logit-transformed influenza activity 1 week after (Table 1). Meanwhile, AH and T are also negatively associated with influenza activity at lag 1 and lag 2, respectively (Table 1).

## Discussion

This study made use of the weekly time series data of influenza and environmental variables in the states of USA during 2010–2015 and three distinct methods for dynamic data analysis (namely, CCM, PCMCI+, and GLM) in order to provide more reliable answers to the question on environmental drivers of influenza. Three sets of results consistently demonstrate the negative impact of ambient $O_3$ on influenza activity in the community.

Hitherto, a limited number of population-level studies have examined the relationship between ambient $O_3$ and influenza, and the findings have been mixed. The integrated assessment of $O_3$ by the USEPA[22] cited two references that reported positive associations of $O_3$ and influenza in Hong Kong and Brisbane[23,24], respectively. The report in Hong Kong, however, was not actually on the relation of $O_3$ and influenza specifically; rather, that report aggregated influenza and pneumonia into one group, which was associated with environmental $O_3$. The other report on the positive association of $O_3$ with pediatric influenza in Brisbane did not strenuously control for potential temporal confounding in the time-series analysis. A more recent time series analysis using Hong Kong surveillance data during 1998–2013 demonstrated that ambient $O_3$ is negatively associated with reduced influenza transmissibility (i.e., real-time effective reproduction number, Rt)[8]. The current study based on the data in the USA, demonstrates again an inhibiting effect of $O_3$ on community-level influenza activity.

One explanation of the $O_3$ inhibition effect on influenza could relate to its direct virucidal potential. $O_3$ inactivation of influenza virus has been reported in studies in vitro. Influenza virus (WSN strain) suspended on a thin film was inactivated within a few hours by an $O_3$ concentration of 160 ppm ($342 \, \mu g/m^3$)[25]. In mice studies, however, $O_3$ exposure at 500 ppb appeared to have no effect on pulmonary virus

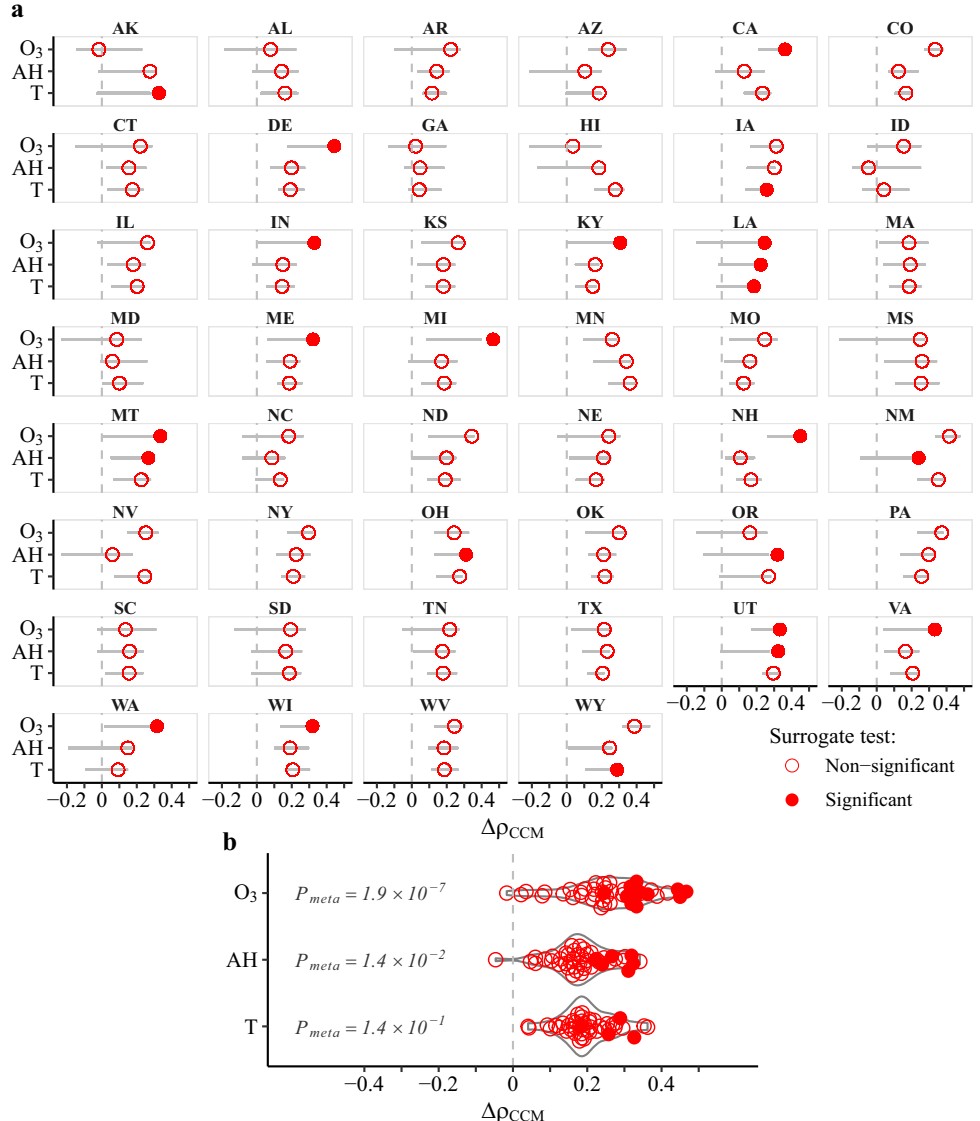

**Fig. 2 | Causality tests by convergent cross-mapping (CCM) for the effect of environmental factors (ozone [O₃], absolute humidity [AH], temperature [T]) at 1-week lag on influenza activity in the states of USA. a** State-specific observed CCM skills (as circles), $\Delta\rho_{CCM}$, and their null distribution in 1000 seasonal surrogates (as line ranges). Circles are filled to signify the measured $\Delta\rho_{CCM}$ for each state exceeding 95% of its null values. **b** Summary of state-specific $\Delta\rho_{CCM}$ values in violin plots. Meta-significance estimate for the nation ($p_{meta}$) is tested by summing the logs of state-level $p$ values; CCM causality is deemed significant with $p_{meta} < 1.0 \times 10^{-3}$.

titers; rather, O₃ diminishes the severity of influenza virus infection and lung injury evidenced by less widespread infection in the lung parenchyma[26,27]. In the current study, the average level of daily maximum 8-h O₃ is less than 40 ppb in the USA. At this low ambient level of O₃, O₃-primed host immunity against influenza infection constitutes a more plausible explanation of the population findings presented here.

Inhaled ambient O₃ primes the pulmonary immune system boosting allergic responses in healthy and susceptible populations[28,29]. Following O₃ exposure, a myriad of immune responses is triggered, and multiple interleukins (IL) are released from epithelial cells, macrophages, and other myeloid cells[30]. Among them, IL-33, acting as an endogenous "alarmin" in response to airway barrier damage incurred by O₃[31,32], is endowed with pleiotropic and homeostatic functions orchestrating airway injury and repair[12,30]. Likewise, IL-33 is also highly expressed following invasion of influenza virus, playing a pivotal role of dynamic immune modulator during the course of infection[33,34]. It is plausible that O₃-induced IL-33 in the cytokine milieu is involved in an immune crosstalk assisting human defense against influenza.

In the setting of inflammation combating foreign antigen, over-expressed IL-33, signaled via its receptor ST2, can be redirected from the default type 2-inducing capacity to augment type 1 immunity, amplifying antiviral CD8+ T cell and natural killer cell responses[13,35,36]. In mice models of influenza infection, exogenous IL-33 inoculation could enhance recruitment of dendritic cells (DCs), increase secretion of pro-inflammatory cytokine IL-12, and prime cytotoxic T-Cell responses, facilitating viral clearance[37]. IL-33 may protect against influenza by orchestrating Th1/Th2 paradigm and so maintaining a fine balance of pro-inflammatory pathogen clearance and anti-inflammatory tissue repair[38,39]. During the resolution phase of infection event, IL-33 acts on residential ST2-expressing group 2 innate lymphoid cells (ILC2s) as well as regulatory T (Treg) cells to restore airway tissue homeostasis, mediated at least partly by amphiregulin (AREG)-dependent repair of virus-damaged epithelium[40,41].

The hypothesis of O₃-elicited IL-33 conferring cross-protection against influenza gains strength further from the evidence of its promising role as a mucosal vaccine adjuvant[42]. Exogenous IL-33 co-

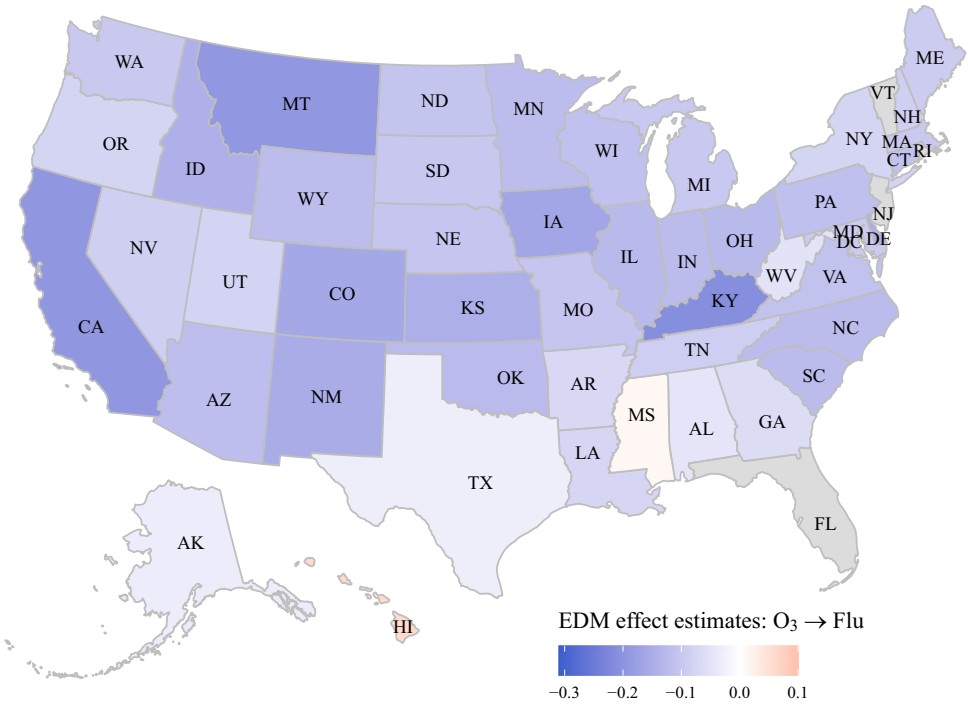

**Fig. 3 | State-specific effect strength estimates for ozone ($O_3$) affecting influenza activity at 1-week lag.** Multivariate S-map technique is used to estimate the effect size, along with CCM test on causality. With normalization of data, the effect magnitude is in a standardized metric. A diverging palette centered at 0 is used to distinguish between positive (red) and negative values (blue). The darker the color, the stronger the effect size in either direction. Four states (FL, NJ, RI, and VT) shaded in gray are excluded from analysis due to influenza data missingness. Map was plotted using "usmap" R package (version 0.6.1)[60].

administered intranasally with recombinant influenza A hemagglutinin (rHA) induced significantly higher antigen (Ag)-specific plasma immunoglobulin G (IgG) and mucosal IgA antibody (Ab) levels as well as enhanced production of both Th1 and Th2-related cytokines, all of which resulted in better protective capacity of the vaccine[43]. Besides, endogenous IL-33 release, within 24 h, after administration of alum-adjuvanted nasal influenza vaccine, induced higher IgA Ab production via enhancing Ag presentation on DCs and promoting ILC2 activation[44]. These findings allude to possible parallels between adjuvanticity of nasally administered alum and ambient $O_3$ exposure.

This study strives to triangulate the evidence by integrating results from different research approaches with distinct theories and working hypotheses. Under the combo methodological framework, CCM employs the idea of SSR for attractors of deterministic dynamical systems thereby addressing better nonlinear state-dependent couplings, but it is less well suited for time series of purely stochastic nature that is better handled with PCMCI+ method[45]. By contrast, PCMCI+ builds on assessing (conditional) probability distribution of random variables, and thus lacks power to recover the non-separable couplings in deterministic systems[46]. On the other hand, the difficulty of PCMCI+ in handling latent (hidden) variables can be addressed in the state-space based method of CCM[47]. The qualitative dependency structure revealed by PCMCI+ can then be complemented by quantitative risk estimates in the time series regression method of GLM by properly controlling for confounding factors.

A few limitations and caveats of the current study require due consideration. The first concern derives from its reliance on passive surveillance data, which can be subject to measurement error. For example, surveillance practices to identify laboratory-confirmed influenza cases as well as healthcare-seeking behaviors due to influenza-like-illness (ILI) can vary across states and years, which may hamper our ability to accurately estimate the actual influenza activity. However, by utilizing a comprehensive proxy measure that combines the laboratory and clinical data, such issue has been minimized to our best effort[48]. Secondly, with the scarcity of virologically confirmed subtype data, we aggregated influenza cases across all influenza subtypes to reduce the number of missing values. Since lineage-specific differences might exist in the effects of $O_3$ and other climatic factors, future studies are warranted to integrate the subtyping and antigenic information into analysis. Thirdly, note that our findings are generated from publicly available state-level weekly data over a time span of 5 years. It remains an important topic for future studies to decode the nuanced relationships of environmental variables with influenza activity on finer spatial and temporal scale, since factors such as demographic features, social connectivity, tourism activities (e.g., Hawaii), as well as public health interventions can lead to fundamentally different base transmission potential, which may interact with environmental factors to shape the complex influenza dynamics[4,49].

In closing, this study reveals a negative impact of ambient $O_3$ on community-level influenza activity by triangulating evidence derived from distinct data analysis approaches. Our finding warrants more laboratory or molecular studies to corroborate the mechanisms shaping the observed causal link in the population, so as to better inform environmental management for public health protection. Moreover, we hope that this work, through a novel integration of divergent analytical frameworks, will catalyze further coordinated efforts in causal discovery using observational dynamic data.

## Methods
### Influenza data
We retrieved state-level weekly laboratory confirmed influenza data from the USA Center for Disease Control and Prevention (CDC) website during the period from October 3, 2010 to September 27, 2015. The counts of laboratory positive cases for influenza (by type A and type B) were reported weekly by designated laboratories located in each state through the platform of World Health Organization (WHO)/ National Respiratory and Enteric Virus Surveillance System (NREVSS)

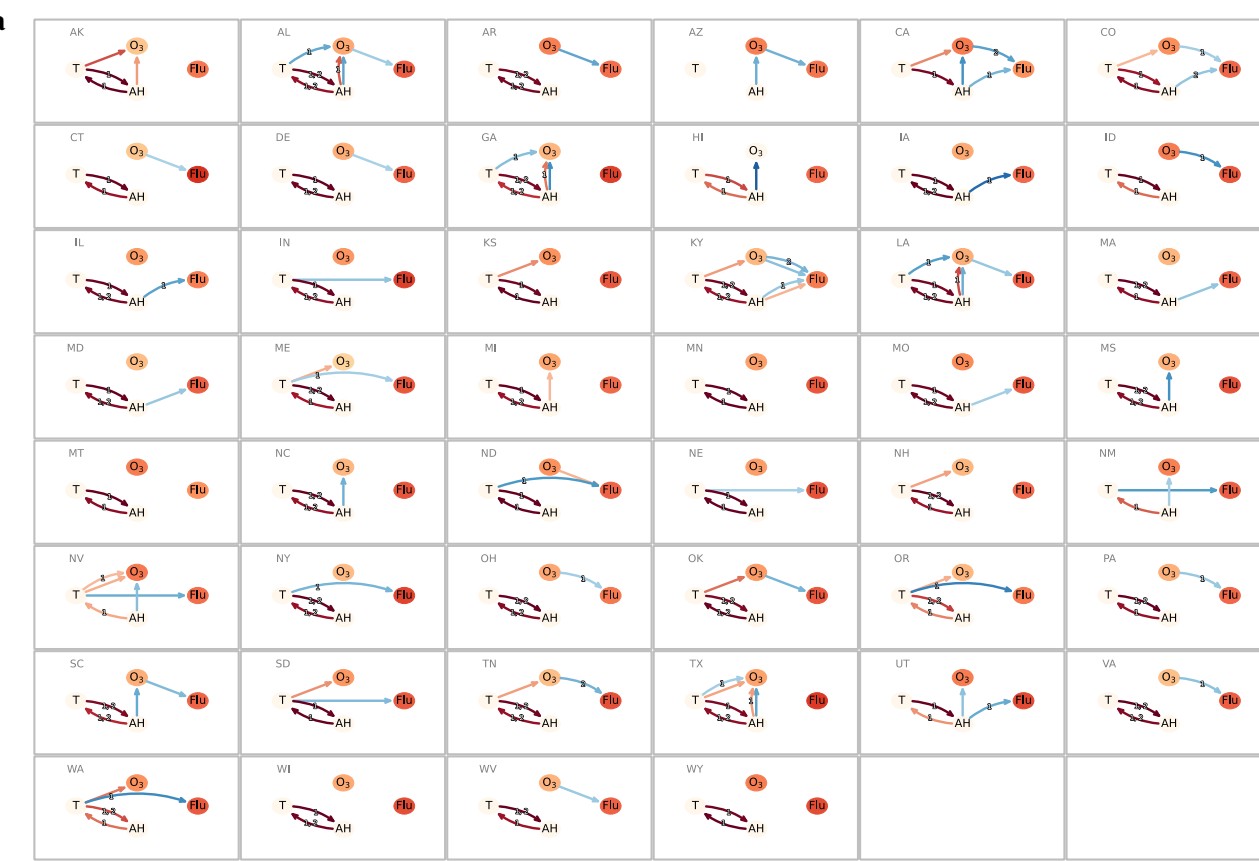

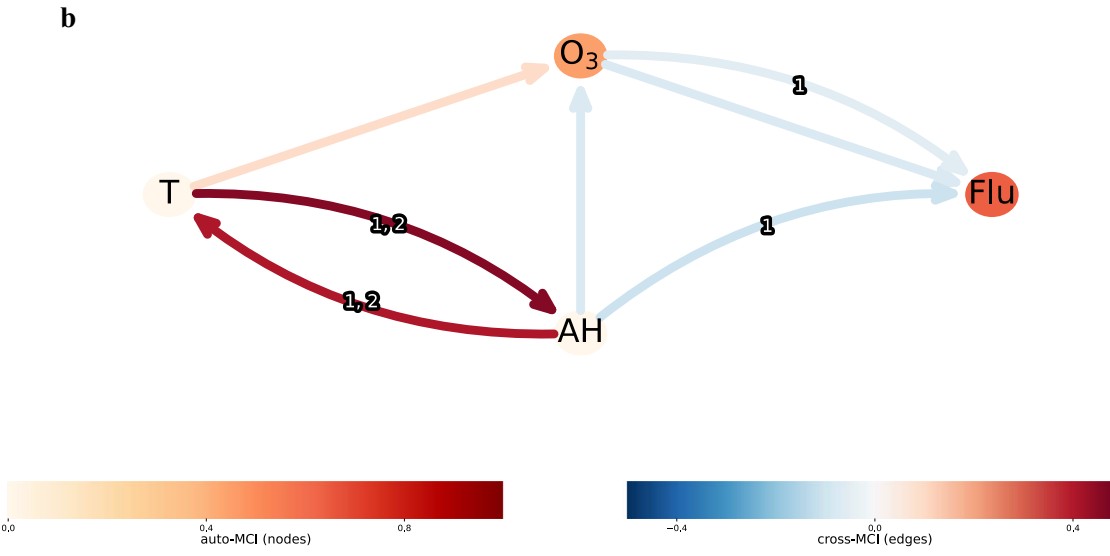

**Fig. 4 | Graphical modeling by PCMCI+ based on dynamic data of environmental factors (ozone [O₃], absolute humidity [AH], temperature [T]) and influenza activity (Flu) in the states of USA. a** State-specific causal graph estimates. Curved and straight edges represent the lagged and contemporaneous causal dependencies, respectively; the number on the curve indicates a lagged relationship in weeks. Node color denotes autocorrelation strength (i.e., auto-MCI [Momentary Conditional Independence] value); edge color depicts the causal strength (i.e., cross-MCI) estimated via partial correlation. **b** Nationwide causal graph estimate. The hyperparameter significance level ($\alpha_{PC}$) is set as 0.05 for individual states and 0.001 for the nationwide analysis.

Collaborating Labs. We also retrieved the weekly reported data on medically attended visits for ILI from CDC for each state during our study period. Over 3000 outpatient healthcare providers throughout the country reported the number of patients with ILI and number of total patient consultations via the U.S. Outpatient Influenza-like Illness Surveillance Network (ILINet) each week.

Due to limited testing capacity in each state, laboratory surveillance data are unable to fully represent influenza activity in the population. Fortunately, ILI data from sentinel outpatient clinics possibly cover a wider spectrum of community influenza cases, despite lower diagnostic specificity. In our study, we incorporated the information from these two available sources to quantify the community-

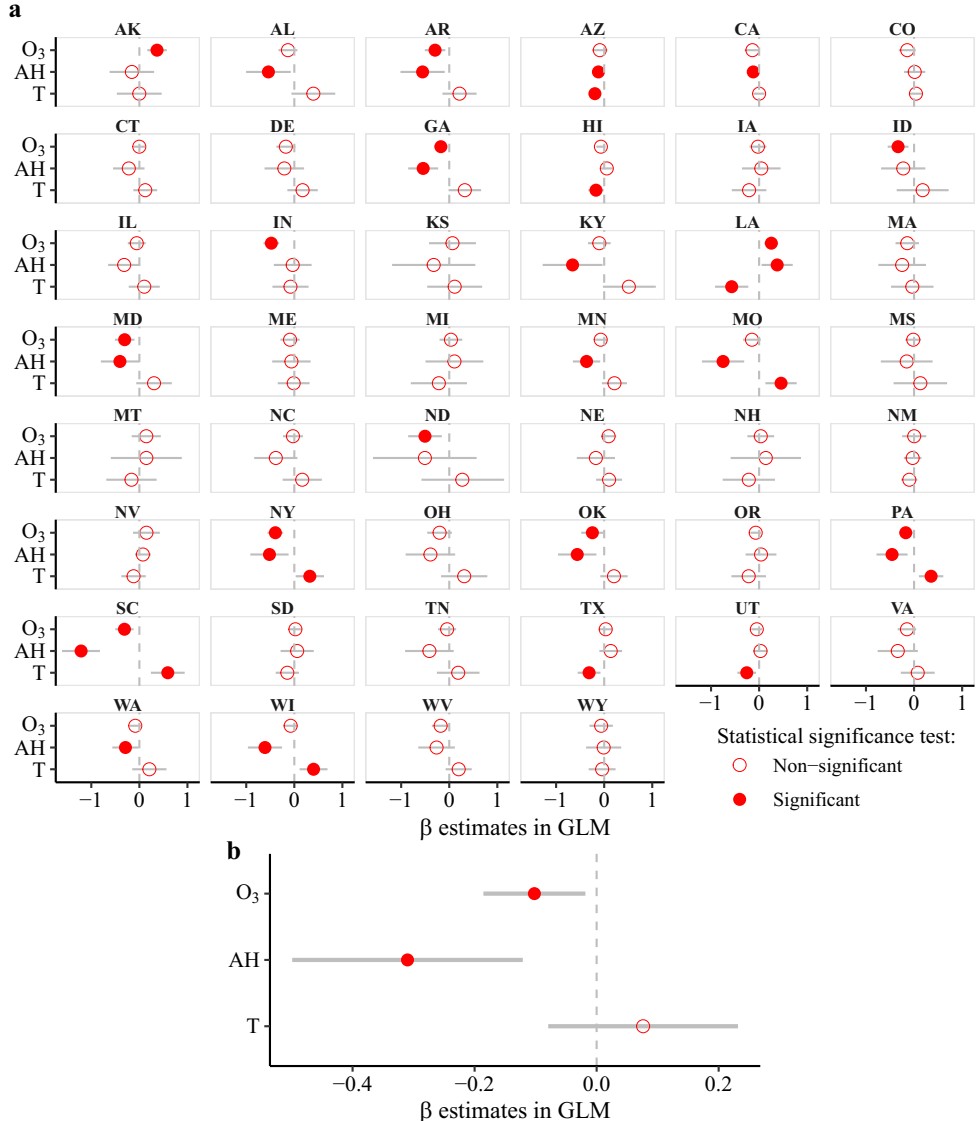

**Fig. 5 | Effects of environmental factors on influenza activity, at 1-week lag, estimated by generalized linear model (GLM). a** State-level point estimates of regression coefficients ($\beta$) and their 95% confidence intervals shown as circles and line ranges, respectively ($n = 173$ weeks of dynamic data); Circles are filled when the $p$ value for statistical significance test is <0.05 (two-sided). **b** Nationwide point risk level influenza activities. estimates with the corresponding 99.9% confidence intervals shown as circles and line ranges, respectively, after pooling state-level coefficients ($n = 46$). Circles are filled when the $p$ value is <0.001 (two-sided) during meta-analysis. $O_3$ ozone, AH absolute humidity, T temperature.

We computed the representative proxy of influenza activity (denoted as "Flu") in community by multiplying the weekly proportion of ILI outpatient consultations with the weekly proportion of specimens tested positive for influenza. That is, $\text{Flu}_{proxy} = (\text{Flu}_{positive}/\text{Flu}_{total})*(\text{ILI}_{outpatient}/\text{Outpatient}_{total})$. This proxy measure, developed and used by epidemiologists to minimize underreporting bias in the laboratory-confirmed influenza data as well as to address unobservability of infection in the ILI data, has been validated as a better linear correlate of influenza virus infections in the community population[50–52].

## Environmental variables data

To explore the proposed environmental drivers of influenza activity, we obtained ambient $O_3$ concentration data in the USA for each state from the Tropospheric Ozone Assessment Report (TOAR) of the International Global Atmospheric Chemistry (IGAC)[53]. Specifically, station-based daily maximum 8-h average $O_3$ levels (ppb) were extracted and averaged by state for analysis. State-level weather data were collected from the National Center for Environmental Information (NCEI), the National Oceanic and Atmospheric Administration (NOAA). During the extraction of environmental variables, a centering approach was used to reduce the station-related measurement bias[54]. That is, raw environmental measurements were centered with respect to their long-term station-specific average and then added with the state-wide average. The daily averages of air temperature (°F) and dew point temperature (°F), computed from hourly land-based station observations, were extracted and converted to Celsius values. Daily absolute humidity, a measure of water vapor density in the air (g/m$^3$), was calculated from the dew point and air temperature, following standard meteorological formulas[55]. The daily $O_3$ and weather time series were further aggregated into week to match the resolution of influenza data.

## Statistical analyses

There are many long stretches of 0 values in the influenza time series, especially outside of flu season, which contain little or very limited

causal information about the environmental factors. Consequently, this study only focused on the flu season (that is considered October through May) in the USA for analysis[56]. States which have at least 3 consecutive years of available influenza data are included. Out of the 50 states, 46 were finally eligible for study; states of Vermont, Rhode Island, New Jersey, and Florida were excluded.

The analytical framework in this study consists of three distinct but complementary methods: CCM, PCMCI+, and GLM, to strive for detection of causal links notably by evidence accumulation (Please refer to a tabulated summary of these three methods in Supplementary Table S1). Firstly, the SSR-based CCM method was applied to examine the existence and strength of causal imprint (as defined in dynamical systems) of candidate drivers on influenza activity, followed by quantification of the effect size using S-map. Secondly, graphical modeling method PCMCI+ was conducted to depict the causal dependency structure (detailed assumptions are explained below and in Table S1) between environmental variables and influenza activity by a DAG[57]. Finally, traditional time series regression GLM was applied to estimate the statistical association of each environmental variable with influenza activity while controlling for the other two simultaneously. To explore the temporal structure of dependency, we tested multiple time lags from 0 to 2 with weekly data. Our analyses were conducted in the R (version 4.1.1) and Python (version 3.8). The "rEDM" package (version 1.9.2)[58], a collection of methods for EDM, was utilized to generate the CCM and S-map results. The "Tigramite" package (version 4.2) was harnessed to complete the PCMCI+ dependency network modeling[17]. The "mgcv" (version 1.8-37)[59] was adopted to fit the quasi-Binomial regression model. Nationwide map for visualizing state-specific S-map effect estimates was produced using "usmap" R package (version 0.6.1)[60].

## CCM

Given dynamic data, CCM which harnesses the technique of SSR for dynamical systems[47] was introduced to evaluate the potential driving role of ambient $O_3$, AH and T on the influenza activity. In the dynamical system theory, as the system state evolves over time, its motion can trace out a trajectory, which collectively form a geometric object often called "attractor manifold", in the multi-dimensional coordinate space whose axes are the set of causally relevant variables such as humidity, $O_3$ concentrations, infection rates, and so forth. Therefore, time series data of observed variables can be simply comprehended as projection of the whole system dynamics onto certain axis. This underpins the SSR technique following the basic tenet of Takens' Theorem[61]; that is, the original multivariate manifold ($M$) can be reconstructed using just one of the system variables (e.g., $X_t$), by taking its delayed coordinates (i.e., embedding) with a time lag $\tau$: $<X_{(t)}, X_{(t-\tau)}, X_{(t-2\tau)}, \ldots, X_{(t-(E-1)\tau)}>$, where $E$ is the embedding dimension that can sufficiently "unfold" the dynamics of system manifold so that reconstructed states on shadow manifold $M_X$ map 1:1 to the original states on $M$[15].

As one of the corollaries to Takens' Theorem, CCM is the causality test kit (from the aspect of dynamical systems) in the EDM framework proposed by Sugihara's research group in 2012[15]. This method assumes low-dimensional deterministic system with limited stochasticity[45]. The basic idea is that if variable $X$ has a causal influence on variable $Y$, then the driven time series $Y_t$ with enough delayed embedding (i.e., reconstructed manifold $M_Y$) should contain the necessary dynamics information to recover or cross-predict the current values of $X_t$, but not vice versa. This practice of using the response variable $Y$ to forecast the causative variable $X$ seems counterintuitive, but it has been well illustrated with algebraic equations by Sugihara's group[15]. The underpinning algorithms of CCM are built on simplex projection[62]. Given the shadow manifold $M_Y$, the $E+1$ nearest neighbors of $y_t$ which correspond to similar system states sharing evolving patterns are first selected. Next, the time indices of these neighboring points of $y_t$ are adopted to locate the corresponding points in $M_X$ (a putative

neighborhood of the predictee). Then, a locally weighted average of the $E+1$ values of $X$ is calculated to predict the cross-mapped estimate of $\hat{x}_t$. Here, the weights are assigned based on the Euclidian distance from $y_t$ to its each nearest neighbor on $M_Y$. The value of $E$ was chosen over the range of 2 to 6 where the maximum of univariate predictability is achieved via leave-one-out cross-validation (Table S2). The lower limit of $E=2$ was specified in order to embed at least one external variable to reconstruct multivariate manifold; the upper limit of $E=6$ was specified because the maximum $E$ should not be larger than the square root of the consecutive time series data length[63], which was 35 or 34 weekly data points in influenza season in the current study.

After the cross-mapping is done, we can evaluate the accuracy (i.e., predictability) by the correlation coefficient ($\rho$) between the predicted and observed values of $X$ series. As the number of data points used for prediction (that is, library size, $L$) becomes larger, the reconstructed shadow manifold $M_Y$ will become denser, and the closer nearest neighbors will accordingly lead to lower estimation error (i.e., higher $\rho$). Such behavior is referred to as "convergence" and is generally utilized to distinguish true causality (as defined in dynamical systems) from simple correlations[15]. Here, we compared the cross-mapping skill ($\rho$) obtained by the maximum library (i.e., the whole data length) to that obtained by the minimum library (i.e., $E+2$ data points allowing for simplex projection), and quantified the convergence property of cross-mapping as $\Delta\rho_{CCM} = \rho_{maxLib} - \rho_{minLib}$. In the vein of Deyle et al., shared seasonality of environmental exposures with influenza activity is another ponderable issue in this context[7]. To preclude spurious CCM results as an artifact of mutual seasonal forcing, we generated an ensemble of 1000 surrogates with randomized seasonal anomalies for the putative cause time series[7]. Consequently, a null distribution of CCM skill ($\Delta\rho_{CCM}$) using surrogate time series was formed. As a test of statistical significance, the cross-mapping skill obtained for the original time series should exceed the 95th percentile of the null distribution built by seasonal surrogates (i.e., at the $\alpha < 0.05$ level). Then, the classical Fisher's methods (that is summation of logs of individual $p$ values) was applied for meta-significance test for all the states[64]. To combat the anti-conservativeness of meta-significance $p$ value estimate, we used a stringent significance threshold of $\alpha$ as $1.0 \times 10^{-3}$. We also repeated CCM analysis setting the candidate cause and effect in the nonsensical reverse direction (i.e., to test whether influenza drives environmental factor) to address the concern of potential synchrony yielding spurious covariation[65].

After qualitative causal relationship was tested, multivariate S-map technique, a method also packed in EDM, was used to examine how and to what degree the putative environmental driver (e.g., $O_3$) influences the influenza activity, thereby quantifying the effect size[21]. Unlike simplex projection using just nearest neighbors, S-map procedure uses all available data points (thus "global") in the library to fit local linear regressions at each successive point along the manifold attractor. Through including a nonlinear localization parameter $\theta$, S-map controls the weighting assigned to each point, thereby tuning how strongly the regression is localized to the target states[66]. The $\theta$ was chosen over the range [0.01, 9] that maximizes the univariate S-map forecast performance using leave-one-out cross-validation (Table S2). By taking multivariate embeddings (i.e., using putative causal variable in addition to time-lagged vectors of the effect variable itself) for SSR[16,67], the S-map coefficients could approximate the relevant Jacobian matrix elements (that is, partial derivatives $\partial Flu/\partial Env$) in the dynamical multivariate state space, which ably indicates the dynamically state-dependent effect strength of environmental factors ("Env") on influenza activity ("Flu")[16,21]. To ensure an equal weighting for variables of different scales in the multivariate SSR model, time series are normalized to unit mean and variance before analysis, so the magnitude of EDM effects is in a standardized metric.

## PCMCI+

PCMCI+, a novel conditional independence-based method proposed by Jakob Runge in 2020[17], was employed as a complementary tool to recognize the dependency structure between $O_3$, AH, T and influenza activity under the graphical modeling framework. Leveraging on the classical PC algorithms (named after the developers Peter and Clark)[68,69] reoriented to observational time series[70], PCMCI+ consists of two main phases, namely skeleton learning phase and subsequent orientation phase. Beyond that, PCMCI+ optimized the selection of conditioning set in conditional independence tests and well exploited autocorrelation, which was demonstrated to yield much higher detection power and orientation recall with better-controlled false positives. Below is a more detailed introduction of PCMCI+ in plain terms. For a good review and comprehensive understanding of this method including algorithms behind, please see Runge et al. papers[17,18].

**Skeleton phase.** Initialized with a complete undirected graph ($\mathcal{G}$) where all nodes (variables) are inter-connected, the goal of skeleton phase is to eliminate the spurious links caused by indirect paths and common drivers via iterative conditional independence tests at some significance threshold $\alpha_{PC}$. Here, for the sake of interpretability and analytical tractability, we implemented conditional independence test with Partial Correlation method (i.e., "ParCorr" in the kit) which assumes linear dependencies but also suffices when the non-linear links can be linearly approximated[46]. As a tuning hyperparameter in the condition-selection step, the significance threshold $\alpha_{PC}$ was set as 0.05 for state-by-state analysis, while significance was calibrated at a stringent $1.0 \times 10^{-3}$ level when analyzing the nationwide graphical structure by concatenating all the state-level time series.

In practice, the skeleton edge removal phase is conducted for lagged and contemporaneous conditioning sets separately. To illustrate, given $X_t^j$ representing each variable/node in the system or target graph ($\mathcal{G}$), we firstly test its putative driver/parent (denoted as $\mathcal{P}(X_t^j)$) over all the time-lagged variables $X_{t-\tau}^i$, where $i,j \in \{1, \ldots, N\}$ and $\tau \in \{1, \ldots, \tau_{max}\}$ (here, $\tau_{max}$ was set as 2 based on domain knowledge). In graphical terms, a causal link $X_{t-\tau}^i \rightarrow X_t^j$ will stand if and only if $X_{t-\tau}^i$ and $X_t^j$ are dependent given any set of conditions. Starting with the first iteration ($r = 0$), unconditional (i.e., bi-variate) independence tests are performed for all the pairs $(X_{t-\tau}^i, X_t^j)$ and $X_{t-\tau}^i$ is removed from the hypothetical $\hat{\mathcal{P}}(X_t^j)$ set if the $p$ value cannot pass the significance level $\alpha_{PC}$. In each next iteration ($r \rightarrow r+1$), we first sort the $\hat{\mathcal{P}}(X_t^j)$ obtained from last iteration by the absolute value of test statistic, and then select the strongest $r+1$ parents as the conditions to conduct conditional independence tests for pairs $(X_{t-\tau}^i, X_t^j)$. After another round of screening, the hypothetical $\hat{\mathcal{P}}(X_t^j)$ set for each $X_t^j$ is further narrowed down and the algorithm will finally converge with no more conditions available for test. In this way, we could identify the lagged potential parents of each variable.

Secondly, to identify contemporaneous potential parents of each $X_t^j$, this stage initializes the graph ($\mathcal{G}$) with all contemporaneous variables presumptively linked, together with all lagged dependencies screened from the previous stage. For all the pairs $(X_{t-\tau}^i, X_t^j)$ (here, $\tau \in \{0, \ldots, \tau_{max}\}$ to also examine contemporaneous causal links), momentary conditional independence (MCI) tests are conducted, iterating through all combinations of subset of the contemporaneous conditions (denoted as $\mathcal{S}$). Besides, the sets $\hat{\mathcal{P}}(X_t^j)$ and $\hat{\mathcal{P}}(X_{t-\tau}^i)$ estimated in the previous step are additionally conditioned on, aiming to account for the common drivers, indirect links, and autocorrelation (i.e., paths blocked) with higher detection power and recall. With similar parents-filtering process at each iteration as the previous step, a skeleton of $\mathcal{G}$ with both contemporaneous and lagged links is finally obtained.

**Orientation phase.** After discovery of the skeleton structure, it is necessary to orient the edges on $\mathcal{G}$ to infer directionality of

relationship. Assumptions of causal stationarity (that is, $X_{t-\tau}^i \rightarrow X_t^j$ holds for any $t$) and time-order (i.e., cause always precedes effect) are applied to help constrain certain cases and simplify the orientation task. For a lagged dependency or adjacency, time order reveals the directionality without ambiguity. While for contemporaneous links, the orientation process can be divided into two steps including a collider orientation stage followed by additional PC constraint rules.

Based on the collider rule, for unshielded triple structures $X_{t-\tau}^i \rightarrow X_t^k - X_t^j (\tau > 0)$ and $X_{t-\tau}^i - X_t^k - X_t^j (\tau = 0)$, we firstly conduct MCI test for pair $(X_{t-\tau}^i, X_t^j)$ conditioning on all possible $\mathcal{S}$, together with their recognized lagged parents (see details above). Second, we store the $\mathcal{S}$ when $X_{t-\tau}^i$ is independent of $X_t^j$. Then we calculate the fraction $\pi$ of $\mathcal{S}$ that contains $X_t^k$. Since a collider (i.e., common effect) would falsely open the link between $(X_{t-\tau}^i, X_t^j)$ (which turns to spurious link) if it were conditioned on, $\mathcal{S}$ is not assumed to contain any collider of the pair $(X_{t-\tau}^i, X_t^j)$ when they are conditionally independent given $\mathcal{S}$. Thus, $\pi$ could be an indicator of the possibility that $X_t^k$ is not a collider. Finally, we use the "majority" rule to decide the existence of colliders (see details in this paper[17]). The considered structure is oriented as a triple of collider when $\pi < 0.5$, as unoriented when $\pi > 0.5$, and as ambiguous when $\pi = 0.5$.

We further determine link directions for the remaining contemporaneous links with three complementary rules. Rule #1 (R1) is to avoid "colliders", since all the colliders are assumed to be already recognized in the collider-hunting stage. For all remaining unshielded structure $X_{t-\tau}^i \rightarrow X_t^k - X_t^j$, orient it as a chain. Rule #2 (R2) is to avoid "cycles" (assuming no feedback loops in the system; colloquially, a variable cannot be its own descendant), which is a tacit assumption when drawing DAGs. For multi-path motifs including $X_t^i \rightarrow X_t^k \rightarrow X_t^j$ with $X_t^i - X_t^j$, orient it as $X_t^i \rightarrow X_t^j$. Rule #3 (R3) is to avoid both "colliders" and "cycles". For structures including $X_t^i - X_t^k \rightarrow X_t^j$ and $X_t^i - X_t^l \rightarrow X_t^j$ where pair $(X_t^l, X_t^k)$ is independent (i.e., not linked) but pair $(X_t^i, X_t^j)$ is of an unoriented link, then orient it as $X_t^i \rightarrow X_t^j$.

Finally, under the standard assumptions of Causal Markov, (adjacency) Faithfulness, causal sufficiency, and causal stationarity (see detailed explanations in Table S1)[18,68,71,72], the output of PCMCI+ algorithms can be interpreted as the causal network structure of the system, conveniently depicted by a directed (or partially directed) acyclic graph (DAG) composed of nodes (representing random variables) and directed edges (representing causal relations). Note that contemporaneous links can remain unoriented indicating the Markov equivalence class or due to conflicting orientation rules. The node color denotes the autocorrelation (labeled "auto-MCI"), varying from 0 to 1, at the lag with maximum absolute value. The link color stands for the sign (i.e., negative or positive) and strength of the connection estimated by MCI test (labeled "cross-MCI") varying from −1 to 1. Straight and curved edges represent the contemporaneous and lagged causal links, respectively; if multiple lagged links occur between paired variables, the color of link will embody the strongest one but with numeric labels indicating all significant lags sorted by cross-MCI values.

## GLM

We applied a conventional time-series regression analysis using generalized linear model (GLM) to estimate the association of $O_3$, AH, and T with influenza activity within each state. Since influenza activity is proportion data, a quasi-Binomial link with logit (i.e., $\log(\frac{Y}{1-Y})$) function that is arguably a reasonable choice, was adopted[73]. By analogy with GLM regression, the target variable "Flu" (i.e., influenza activity) was logit-transformed in EDM and PCMCI+ analyses as well, which could give us an overall interpretation benchmark. Besides, due to exponential transmission pattern of influenza cases, such scale transformation can help discern small differences in influenza activity. To modulate the case when $Y$ takes a value of 0, a random small number (i.e., 25% to 100% of non-zero minimum "Flu" level for each specific influenza season) was added to allow for defined transformation. To filter out the potential

confounding effect of unmeasured variables, we included dummy variables for the year to capture the secular trend, and dummy variables for each month of a year to capture seasonality in the model. To account for strong autocorrelation caused by disease transmission, we took the logarithm of 1-week lagged outcome variable (i.e., $\log(Y_{t-1})$) as another covariate in the model, which was provably able to match the likely mechanism better (so named "transmission term") and predict outcomes with reduced residual dispersion[74].

When estimating the relationship of $O_3$ with influenza, the same-week AH and T are simultaneously included in the model as a linear term to control for confounding. The regression coefficient estimates together with their corresponding 95% confidence intervals (CIs) were computed for exposures at lag 0, lag 1, and lag 2 (in weeks), respectively, for each state. The state-wise and lag-specific effect estimates were pooled with a random-effects meta-analysis (using restricted maximum-likelihood estimator for the between-study variance)[75], with the statistical significance threshold set stringently as $1.0 \times 10^{-3}$.

### Reporting summary
Further information on research design is available in the Nature Portfolio Reporting Summary linked to this article.

## Data availability
Th raw data on influenza and ILI are publicly available at https://www.cdc.gov/flu/weekly/index.htm. The ozone data used in this study are publicly available at http://www.igacproject.org/activities/TOAR. The climate data used in this study are publicly available at https://www.ncdc.noaa.gov/data-access/land-based-station-data/land-based-datasets. The data set supporting the findings of this work is available at the GitHub repository[76].

## Code availability
The codes for reproducing our results can be found at the GitHub repository[76]: https://zenodo.org/records/10892898. A step-by-step demonstration on data analysis is also provided in the Supplementary Materials, to facilitate understanding in a structured manner.

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

## Acknowledgements

This work was supported by the research funding to L.T. as listed below: the Health and Medical Research Fund (HMRF) from the Food and Health Bureau (FHB) of Hong Kong (Ref. No.: 18192061; 20211551), the General Research Fund (GRF) scheme of the Hong Kong Research Grants Council (Ref. No.: 17613819; 17115122; 17109723), the Theme-based Research Scheme (TRS) of the Hong Kong Research Grants Council (Ref. No.: T24-508/22-N), the General Program of National Natural Science Foundation of China (Ref. No.: 82173469), the Guangdong Natural Science Fund (GD-NSF) of China (Ref. No.: 2022A1515011151), and Seed Funding for Strategic Interdisciplinary Research Scheme from the University Research Committee at the University of Hong Kong (Ref. No.: 102010191). Data cleaning and management was assisted by Mr. King Pang Chan. We thank Dr. Andreas Gerhardus for discussions on the application of PCMCI+. The computations were performed using research computing facilities offered by Information Technology Services, the University of Hong Kong.

## Author contributions

L.T., F.G., and V.D. designed research; L.T., F.G., P.Z., and V.D. performed research; F.G., P.Z., Z.H., S.D., and H.L. collected data; L.T., F.G., P.Z., and

V.D. analyzed data; F.G., P.Z., V.D., and L.T. wrote the paper; Y.G., J.R., and K.Z. provided methodological guidance; S.T.A., R.C., and Y.G. revised the paper.

## Competing interests

The authors declare no competing interests.
