## [Peer Review File · Nature Communications]

Ozone as an environmental driver of influenzaReviewers' comments:

Reviewer #1 (Remarks to the Author):

The authors use a convergent cross mapping (CCM) to explore the link between ozone and influenza cases in the US. They find evidence that ozone is a better predictor of influenza cases than absolute humidity and temperature. My main concern is I think the authors could do more to convince us of the link between ozone and influenza, beyond applying the CCM model. The manuscript reads more as an exercise in applying CCM methods (and given there is an R-package, this is now quite easy to do) than in trying to demonstrate a clear link between O₃ and influenza. There have been some strong critiques of CCM and its applicability to infectious disease, and as such, I think the authors need to rely on supporting evidence beyond the CCM results. This is especially the cases as there is scant mechanistic evidence (from laboratory studies) linking O₃ to influenza. In summary, I am not convinced by the findings in this paper.

Major comments:

The authors cite an earlier PNAS paper [Deyle et al 2016] that applied this CCM to influenza, finding a link with humidity and temperature. However, the key difference is that there is substantial mechanistic evidence (from laboratory studies) linking temperature and humidity to influenza survival and transmission. I think the authors have more work to do with convincing the reader that ozone is a driver, as there is very little mechanistic evidence that it affects influenza survival. It seems that the three studies the authors cite do not conclusively link influenza and O₃? Is there any other evidence? What is the mechanistic pathway for the ozone effect?

CCM has received some reasonable criticism in terms of its applicability to infectious disease [Baskerville and Cobey 2016,2017]. In particular, it was shown that the approach is unable to distinguish whether the environmental variables drive influenza cases or vice versa: that cases drive environmental variables [Baskerville and Cobey, 2017] (while this may seem like a silly exercise, testing the directionality of an effect is often a critical first assessment of whether a causal inference technique really works). Given the known issues with this approach, is this really the best method for showing the ozone-influenza link? Does the author's model also show that influenza cases drive ozone?

The authors argument for the O₃ driver rests only on the CCM results, but is there other evidence? For instance, we know that influenza dynamics closer to the tropics e.g. Hawaii, tend to be more persistent. Is this the same with ozone seasonality in the tropics? Do spatial patterns of ozone match spatial patterns of influenza? (Notably, Deyle 2016 actually showed some of this spatial evidence first, before applying CCM).

The availability of the R-package for CCM makes it very easy to apply this method without real consideration of the topic at hand. I think the authors could substantially improve this paper by focusing on the question, not solely the methodological approach. It isn't clear to me why the authors chose to do this investigation into ozone, as opposed to another environmental variable. There is much more support for a possible role of precipitation, or even UV radiation, on influenza virus transmission (though a notable lack of population-level studies). Given climate variables are highly correlated, can the CCM model really find a distinct effect of ozone compared to other variables? I would be concerned if, for instance, it found a positive result for any environmental variable. Perhaps the authors can test this.

Why do the authors only analyze data from 28 states? To my knowledge, influenza reporting and surveillance at the state level is very good – shouldn't there be more data? (I just checked and could see flu data from Hawaii and Maine at least). Are states with "missing" values because cases are in fact zero? Also, why do the authors only take data from 2010-2015, when there is data available up until the present. Wouldn't this increase the power of their study?

Can the authors run any other basic statistical tests to demonstrate the ozone link? For instance, can you show that states with higher ozone have lower transmission. Or that a year with particularly high ozone, has lower transmission. One way to do this would be to use a simple

regression model. The authors could calculate transmission for each point of time and location in the dataset, put transmission on the left-hand side and ozone on the right, with controls for time and location.

Line 98-99: if we were looking at the effect of climate on transmission, wouldn't we expect it to be static over time? It seems a weakness of CCM that it can't capture this? Similarly for Figure , what does it mean that the effect size varies across space? Why would we expect this to vary? It doesn't seem to vary in a way that corresponds to either influenza or ozone – is this just a failure of the ozone model to capture spatial differences?

Line 144; Again, I don't quite understand why the O3 effect estimates vary over time –does this mean that if $\text{influenza} = \beta * \text{ozone}$, this β is varying, or do you just mean that there is a constant effect of ozone on influenza but ozone varies seasonally? I guess it isn't clear if the β estimate is varying or if it's just ozone. If I assessed the influence of a completely random variable on influenza, I could make the argument it drove influenza by saying that the effect estimate varied over time and space.

Minor comments:

Line 53-55. I think the authors could better summarize these studies. I'm not sure what "owns virucidal action" means? It seems the evidence linking ozone to influenza is somewhat weak here?

Fig 2. I'm not clear on what the "0" line represents here. Improvement compared to what baseline scenario? I think this could be better explained in the text (lines 82 -93).

Reviewer #2 (Remarks to the Author):

The authors discuss existing laboratory and clinical evidence for effects of ozone (O3) on influenza infections, motivating their population-level investigation of a causal relationship. While the analysis only spans states within the continental U.S., and thus is only looking at seasonal temperate climates, the authors note previous population-level evidence in Hong Kong. Thus, the study is certainly of sufficient scope and scholarly merit for publication. The analysis are presented in a clear, step-by-step manner and the authors have done due diligence in including code and data to enable close examination and reproduction of the calculations and results.

The concerns I have may well be easily resolved, but I do have a major concern about some of the details of Figure 1 that I believe point to an error or inconsistency in a piece of the calculations (which I provide detail on below).

General questions:

- Why is the analysis focused on 2010-2015? Is some portion of the data not available earlier?
- Does O3 particularly help explain the deviations of influenza from seasonality? (My understanding is that O3 has a less pronounced seasonal signature in most or all states than T or AH).

Specific questions/concerns:

Lines 102-103: the term "unit-free" leaves some ambiguity here. My understanding from other parts of the manuscript is that these are relative to the standard variation of the observed dynamics of those variables, but I think it would be good to state that here for clarity.

Lines 142-145: I'm not sure this argument is entirely correct. The inhibiting effect of O3 would be a constant for an individual not the population, no? but shouldn't be expected at the population level but rather should depend on the current number of cases. If there are 0 cases of influenza, then a change in O3 should have no effect on the cases next week. Indeed, the fact that it shows a highest effect in winter may simply be just because cases are high. Doing an appropriate normalization might be tricky, though, since the cases time series contain 0 counts.

Line 303: An S-map parameter of 0.09 is stated, which would represent fairly weak nonlinearity. However, looking into the SI, it seems this probably supposed to read "0.9" not "0.09".

Figure 2: It is not clear how the results are being displayed. My best guess is that these are standard box-and-whisker plots with boxes on 25th and 75th quantiles across the samples in each group, and that each sample represents the measured forecast improvement for a single state included in the analysis. The methods relevant to forecast improvement are also ambiguous. Based on the example calculations shown in the SI I am inferring each state was analyzed separately. In both cases, it should be clearly stated in the caption and methods. Additionally, if the analysis was done on a state-by-state basis, it is odd to me that the state-by-state results are shown for CCM and not forecast improvement. I'd request these be included in the supplement.

Methods. My thanks to the authors for including markdown of core calculations in the supplement. Unfortunately, it did not appear that the `dat_wi.rda`, `analysis_wi.R`, and `functions_wi.R` were actually included in the manuscript files available to me. Nevertheless, the markdown code and `xlsx` files that were available were quite helpful.

In reviewing the markdown, one piece of the analysis did stick out, which was how the univariate embedding dimension is determined with out-of-sample predictability, splitting the time series so that the first 2/3 are in-sample training set and the last 1/3 are the out-of-sample test set for predictions. Given that none of the other calculations appear to be done with the same out-of-sample split (including the optimal E calculations for CCM), it seems like an odd choice that will just increase the variance in the results. Determining E on a relatively small prediction set seems like a good way to under embed the system. Unless there is a strong justification I'm missing, I'd strongly recommend that the authors simply use leave-one-out cross-validation for determining E as they do with the remainder of the EDM calculations. This should give a more robust result that should be expected to hold up better e.g. to subsequent testing on newer data.

Additionally, I was unable to find the exact calculation for obtaining the proxy measure of influence (main text lines 203-213) from the data contained in the supplemental `*.xlsx` file. That file contains "flu", "flu_total", "ILI", and "ILI_total". My best interpretation of the text is that $\text{fluP} = (\text{flu}/\text{flu_total}) * (\text{ILI}/\text{ILI_total})$.

This brings me to my biggest question about the main findings. Several details of what is shown in Fig. 1 were a bit surprising to me so I did some digging to try to understand better. In particular, I was very surprised to see in Maryland and Delaware that the seasonal surrogates showed a distribution centered on 0. This surprises me for two reasons. (1) My expectation that influenza and climate in these states is markedly seasonal. Indeed, when I looked at the data in the `.xlsx` file the linear correlations between "AH" and "flu", or "AH" and "fluP" (as calculated by my best guess described above) definitely reflect this for Delaware $\text{cor}(\text{AH}, \text{flu}) = -0.48$; $\text{cor}(\text{AH}, \text{fluP}) = -0.42$. (2) My other expectation is that influenza and climate are strongly correlated to nearby states that show very different surrogate results, namely New York, Pennsylvania, West Virginia, and Virginia. Indeed checking correlations, the climate variables have extremely high correlations ($\text{cor}(\text{AH_Delaware}, \text{AH_NewYork}) = 0.99$), and the influenza high as well ($\text{cor}(\text{fluP_Delaware}, \text{fluP_NewYork}) = 0.80$).

Given all this, I have to believe there is some sort of error or inconsistency in the surrogate calculations. My immediate guess is that it is related to missing data in the time series. I notice that Delaware and Maryland are missing 42 and 39 points, respectively, meaning they are amongst the gappiest of the states. For that matter, South Carolina, Montana, Louisiana, and Oregon all also show very low surrogate `rho_ccm` and are the other states with a large number of missing values. From the markdown it is not clear to me how missing values are handled with the surrogate procedure and the EDM calculations themselves, so that is as far as I can follow my guess. Not having access to state-by-state "forecast improvement" results, it's hard for me to assess if there might be similar artifacts in that piece of the analysis as well (motivation for my request above to include those at least in the supplement). I would ask the authors check that carefully. That said, it seems like if there are issues it is just with a few of the states, so I would guess that the overall results of the paper won't significantly change.

Reviewer #3 (Remarks to the Author):

This paper examines the role of ozone on the transmission of influenza in the continental United States. The authors employ causal framework through EDM modeling of environmental variables and state-level influenza data.

The authors thoroughly explain a complex modelling system and provide evidence of the involvement of ozone in the causal pathway for population-level influenza cases.

Can the authors provide a causal diagram to illustrate the role of ozone in the causal pathway? Given that temperature influences the amount of ozone and also influenza, is temperature a mediator or a confounder in the relationship?

Taking a state-wide average of environmental variables has some limitations - there is certainly a lot of within-state variability that is lost by doing this. Can the authors comment on this limitation and the ability of their method to account for this? Would we expect data at a finer spatial resolution to give similar results?

Minor edits

-line 54: "exhibits" might be better than "owns" in this sentence

-line 205: remove "should"

Regards,

Rachel Sippy

Kindly note that the texts in **black** are the comments from the reviewers, and our responses are positioned right after the comments and colored in **blue**. The changes made in the revised manuscript following the reviewers' comments and suggestions are also **highlighted**.

Reviewer #1 (Remarks to the Author):

The authors use a convergent cross mapping (CCM) to explore the link between ozone and influenza cases in the US. They find evidence that ozone is a better predictor of influenza cases than absolute humidity and temperature. My main concern is I think the authors could do more to convince us of the link between ozone and influenza, beyond applying the CCM model. The manuscript reads more as an exercise in applying CCM methods (and given there is an R-package, this is now quite easy to do) than in trying to demonstrate a clear link between O₃ and influenza. There have been some strong critiques of CCM and its applicability to infectious disease, and as such, I think the authors need to rely on supporting evidence beyond the CCM results. This is especially the cases as there is scant mechanistic evidence (from laboratory studies) linking O₃ to influenza. In summary, I am not convinced by the findings in this paper.

Re: We appreciate very much the critical and constructive comments which have propelled us to re-do the analysis and re-write the paper. Two major criticisms were: 1) findings based on one single method of CCM, and 2) scant mechanistic evidence for a link between O₃ and influenza. Now we use 3 methods with disparate theoretical assumptions and hidden biases which complement each other to demonstrate the consistent linkage between O₃ and influenza supported by the latest mechanistic evidence in immunology literature.

Major comments:

1. The authors cite an earlier PNAS paper [Deyle et al 2016] that applied this CCM to influenza, finding a link with humidity and temperature. However, the key difference is that there is substantial mechanistic evidence (from laboratory studies) linking temperature and humidity to influenza survival and transmission. I think the authors have more work to do with convincing the reader that ozone is a driver, as there is very little mechanistic evidence that it affects influenza survival. It seems that the three studies the authors cite do not conclusively link influenza and O₃? Is there any other evidence? What is the mechanistic pathway for the ozone effect?

Re: The mechanistic pathway for the ozone effect on influenza is being pieced together thanks to the rapid progress in immunological findings of ozone, interleukin-33 (IL-33), and influenza.

- 1) Ozone exposure induces IL-33 in airways, which is an early defense mechanism in physiological conditions.
- 2) IL-33 protects against influenza by: a) limiting infection; b) boosting tissue repair during infection recovery; and c) augmenting mucosal vaccine efficacy as an adjuvant.

Ozone exposure induces IL-33 in airways. Ozone exposure has been found to induce IL-33

mRNA activation and increase IL-33 protein expression in mice airways [Ref: Yang_2016]. Upon ozone exposure and immediate lung barrier injury, the alarmin IL-33 is rapidly released from epithelial cells, macrophages, and other myeloid cells, which have homeostatic functions in that lung injury and inflammation are enhanced in the absence of IL-33 [Ref: Michaudel_2018]. Indeed, multiple interleukins are released upon ozone exposure to regulate inflammation; while IL-1 α might be enhancing the inflammatory response, IL-33 tends to attenuate inflammation and maintain airway epithelial integrity upon the immediate epithelial barrier damage by ozone [Ref: Sokolowska_2019]. It is likely that IL-33 induced by ozone exposure contribute to cross-protection against influenza.

- Yang, Q., Ge, M. Q., Kokalari, B., Redai, I. G., Wang, X., Kemeny, D. M., Bhandoola, A., & Haczku, A. (2016). Group 2 innate lymphoid cells mediate ozone-induced airway inflammation and hyperresponsiveness in mice. *The Journal of Allergy and Clinical Immunology*, 137(2), 571–578. <https://doi.org/10.1016/j.jaci.2015.06.037>
- Michaudel, C., Mackowiak, C., Maillat, I., Fauconnier, L., Akdis, C. A., Sokolowska, M., Dreher, A., Tan, H.-T. T., Quesniaux, V. F., Ryffel, B., & Togbe, D. (09/2018). Ozone exposure induces respiratory barrier biphasic injury and inflammation controlled by IL-33. *The Journal of Allergy and Clinical Immunology*, 142(3), 942–958. <https://doi.org/10.1016/j.jaci.2017.11.044>
- Sokolowska, M., Quesniaux, V. F. J., Akdis, C. A., Chung, K. F., Ryffel, B., & Togbe, D. (2019). Acute Respiratory Barrier Disruption by Ozone Exposure in Mice. *Frontiers in Immunology*, 10, 2169. <https://doi.org/10.3389/fimmu.2019.02169>

IL-33 protection against influenza by limiting the infection. During influenza infection, IL-33 was found to augment protective type 1 immunity by promoting expansion of CD8⁺ T cells [Ref: Bonilla2012]. In mice models of influenza infection, exogenous IL-33 inoculation enhances antiviral immunity by inducing the recruitment of dendritic cells, increasing the secretion of IL-12, and promoting cytotoxic CD8⁺ T cell responses in the local microenvironment [Ref: Kim2019].

- Bonilla, W. V., Fröhlich, A., Senn, K., Kallert, S., Fernandez, M., Johnson, S., Kreutzfeldt, M., Hegazy, A. N., Schrick, C., Fallon, P. G., Klemenz, R., Nakae, S., Adler, H., Merkler, D., Löhning, M., & Pinschewer, D. D. (2012). The Alarmin Interleukin-33 Drives Protective Antiviral CD8⁺ T Cell Responses. *Science*, 335(6071), 984–989. <https://doi.org/10.1126/science.1215418>
- Kim, C. W., Yoo, H. J., Park, J. H., Oh, J. E., & Lee, H. K. (2019). Exogenous Interleukin-33 Contributes to Protective Immunity via Cytotoxic T-Cell Priming against Mucosal Influenza Viral Infection. *Viruses*, 11(9). <https://doi.org/10.3390/v11090840>

IL-33 protection against influenza by boosting tissue repair. During the recovery phase of influenza infection, tissue repair and maintenance is mediated by IL-33-induced production of amphiregulin (AREG), a tissue remodeling-associated protein, by immune cells such as regulatory T cells (Tregs) [Ref: Arpaia_2015] and type 2 innate lymphoid cells (ILC2s) [Ref: Monticelli_2011].

- Monticelli, L. A., Sonnenberg, G. F., Abt, M. C., Alenghat, T., Ziegler, C. G. K., Doering, T. A., Angelosanto, J. M., Laidlaw, B. J., Yang, C. Y., Sathaliyawala, T., Kubota, M., Turner, D., Diamond, J. M., Goldrath, A. W., Farber, D. L., Collman, R. G., Wherry, E. J., & Artis, D. (2011). Innate lymphoid cells promote lung-tissue homeostasis after infection with influenza virus. *Nature Immunology*, 12(11), 1045–1054. <https://doi.org/10.1031/ni.2131>
- Arpaia, N., Green, J. A., Moltedo, B., Arvey, A., Hemmers, S., Yuan, S., Treuting, P. M., & Rudensky, A. Y. (2015). A Distinct Function of Regulatory T Cells in Tissue Protection. *Cell*, 162(5), 1078–1089. <https://doi.org/10.1016/j.cell.2015.08.021>

IL-33 protection against influenza by augmenting mucosal vaccine efficacy as an adjuvant. Intranasal delivery of IL-33 enhanced the efficacy of an inactivated influenza virus vaccine by early activation of lung ILC2 and strongly induced humoral immunity, specifically IgA antibodies in the lung mucosae of mice models [Ref: Williams_2021]. IL-33 release, within 24 h, from necroptotic alveolar epithelial cells upon administering a nasal alum-adjuvanted influenza vaccine led to enhanced IgA production [Ref: Sasaki_2021]. Exogenous IL-33 increased IgG and IgA production in plasma and respiratory mucosa, respectively, upon intranasal administration of recombinant influenza A hemagglutinin (rHA) protein in mice [Ref: Kayamuro_2010]. IL-33 was found to be essential for adjuvant immunogenicity of Hydroxypropyl- β -Cyclodextrin co-administered with intranasal influenza vaccines [Ref: Kobar_2020].

- Williams, C. M., Roy, S., Califano, D., McKenzie, A. N. J., Metzger, D. W., & Furuya, Y. (2021). The Interleukin-33-Group 2 Innate Lymphoid Cell Axis Represents a Potential Adjuvant Target To Increase the Cross-Protective Efficacy of Influenza Vaccine. *Journal of Virology*, 95(22), e0059821. <https://doi.org/10.1128/JVI.00598-21>
- Sasaki, E., Asanuma, H., Momose, H., Furuhashi, K., Mizukami, T., & Hamaguchi, I. (2021). Nasal alum-adjuvanted vaccine promotes IL-33 release from alveolar epithelial cells that elicits IgA production via type 2 immune responses. *PLoS Pathogens*, 17(8), e1009890. <https://doi.org/10.1371/journal.ppat.1009890>
- Kayamuro, H., Yoshioka, Y., Abe, Y., Arita, S., Katayama, K., Nomura, T., Yoshikawa, T., Kubota-Koketsu, R., Ikuta, K., Okamoto, S., Mori, Y., Kunisawa, J., Kiyono, H., Itoh, N., Nagano, K., Kamada, H., Tsutsumi, Y., & Tsunoda, S.-I. (2010). Interleukin-1 Family Cytokines as Mucosal Vaccine Adjuvants for Induction of Protective Immunity against Influenza Virus. *Journal of Virology*, 84(24), 12703–12712. <https://doi.org/10.1128/JVI.01182-10>
- Kobari, S., Kusakabe, T., Momota, M., Shibahara, T., Hayashi, T., Ozasa, K., Morita, H., Matsumoto, K., Saito, H., Ito, S., Kuroda, E., & Ishii, K. J. (2020). IL-33 Is Essential for Adjuvant Effect of Hydroxypropyl- β -Cyclodextrin on the Protective Intranasal Influenza Vaccination. *Frontiers in Immunology*, 11, 360. <https://doi.org/10.3389/fimmu.2020.00360>

2. CCM has received some reasonable criticism in terms of its applicability to infectious disease [Baskerville and Cobey 2016,2017]. In particular, it was shown that the approach is unable to distinguish whether the environmental variables drive influenza cases or vice versa: that cases drive environmental variables [Baskerville and Cobey, 2017] (while this may seem like a silly exercise, testing the directionality of an effect is often a critical first assessment of whether a causal inference technique really works). Given the known issues with this approach, is this really the best method for showing the ozone-influenza link? Does the author’s model also show that influenza cases drive ozone?

Re: No, CCM is not really the single best approach. The CCM method has its own issues. However, on the ‘directionality of an effect’, we do not have CCM results suggesting “influenza cases drive ozone” when setting the candidate cause and effect in reverse. Please see the figure below for details.

Figure S2. Reverse causality test for the effect of influenza intensity, at 1-week lag, on environmental measurements (ozone [O_3], absolute humidity [AH], temperature [T]) by

convergent cross-mapping (CCM). Panel A shows the observed CCM skills (as circles), $\Delta\rho_{CCM}$, and their null distribution tested from 1,000 seasonal surrogates (as line ranges). Panel B shows a summary of state-specific $\Delta\rho_{CCM}$ values in violin plots. Circles are filled to signify the measured $\Delta\rho_{CCM}$ for each state exceeding 95% of its null values. When summing the logs of state-specific P values to obtain the meta-significance estimate for the nation (P_{meta}), none of the 3 environmental factors are found to be driven by influenza.

To strengthen the fidelity of making causal inference, we now apply three distinct methods for data analysis:

- CCM under the general umbrella of EDM (empirical dynamic modelling),
- PCMCI+ (Peter-Clark-momentary-conditional-independence plus, a causal graph-based method), and
- GLM (generalized linear model, a traditional regression method)

The combination of these three methods for causal inference from dynamic data are expected to complement each other because of their distinct conceptual principles. While the CCM method better addresses nonlinear state-dependent couplings, it is less well suited for time series of a stochastic nature that is better handled with PCMCI+ method. On the other hand, the difficulty of PCMCI+ in handling latent (hidden) variables can be partially addressed in the method of CCM. The qualitative dependency structure revealed by PCMCI+ can then be complemented by quantitative risk estimates in the time series regression method of GLM upon properly controlling for confounding factors.

Examples of CCM applications:

- Sugihara, G., May, R., Ye, H., Hsieh, C.-H., Deyle, E., Fogarty, M., & Munch, S. (2012). Detecting causality in complex ecosystems. *Science*, 338(6106), 496–500. <https://doi.org/10.1126/science.1227079>
- Sugihara, G., Deyle, E. R., & Ye, H. (2017). Reply to Baskerville and Cobey: Misconceptions about causation with synchrony and seasonal drivers. *Proceedings of the National Academy of Sciences of the United States of America*, 114(12), E2272–E2274. <https://doi.org/10.1073/pnas.1700998114>
- Deyle, E. R., Maher, M. C., Hernandez, R. D., Basu, S., & Sugihara, G. (2016). Global environmental drivers of influenza. *Proceedings of the National Academy of Sciences*, 113(46), 13081–13086. <https://doi.org/10.1073/pnas.1607747113>
- Ushio, M., Hsieh, C.-H., Masuda, R., Deyle, E. R., Ye, H., Chang, C.-W., Sugihara, G., & Kondoh, M. (2018). Fluctuating interaction network and time-varying stability of a natural fish community. *Nature*, 554(7692), 360–363. <https://doi.org/10.1038/nature25504>

Examples of PCMCI+ applications:

- Runge, J., Nowack, P., Kretschmer, M., Flaxman, S., & Sejdinovic, D. (2019). Detecting and

quantifying causal associations in large nonlinear time series datasets. *Science Advances*, 5(11), eaau4996. <https://doi.org/10.1126/sciadv.aau4996>

- Krich, C., Runge, J., Miralles, D. G., Migliavacca, M., Perez-Priego, O., El-Madany, T., Carrara, A., & Mahecha, M. D. (2020). Estimating causal networks in biosphere–atmosphere interaction with the PCMCI approach. *Biogeosciences*, 17(4), 1033–1061. <https://doi.org/10.5194/bg-17-1033-2020>
- Runge, J., Bathiany, S., Bollt, E., Camps-Valls, G., Coumou, D., Deyle, E., Glymour, C., Kretschmer, M., Mahecha, M. D., Muñoz-Marí, J., van Nes, E. H., Peters, J., Quax, R., Reichstein, M., Scheffer, M., Schölkopf, B., Spirtes, P., Sugihara, G., Sun, J., ... Zscheischler, J. (2019). Inferring causation from time series in Earth system sciences. *Nature Communications*, 10(1), 2553. <https://doi.org/10.1038/s41467-019-10105-3>

3. The authors argument for the O3 driver rests only on the CCM results, but is there other evidence? For instance, we know that influenza dynamics closer to the tropics e.g. Hawaii, tend to be more persistent. Is this the same with ozone seasonality in the tropics? Do spatial patterns of ozone match spatial patterns of influenza? (Notably, Deyle 2016 actually showed some of this spatial evidence first, before applying CCM).

Re: Yes. In our new manuscript, we supplemented another two analytical methods (PCMCI+ and GLM) to decipher the relationship of O₃ with influenza, and a consistent negative dependency has been identified. Besides, mechanistic evidence (from laboratory studies) linking O₃ and influenza has been thoroughly reviewed and is now pieced together, that is ambient O₃ could induce interleukin-33 which in turn protects against influenza.

Deyle et al (2016) paper did not examine the spatial correspondence between ozone and influenza, which is subject to spurious correlation. Rather, they checked on the correspondence between seasonality of environment and seasonality of influenza infection --- they found Spearman correlation between the two was high: $\rho = 0.73$ [For example, the lack of influenza seasonality in the tropics corresponded with the lack of ‘environmental’ seasonality in the tropics]. With this shared seasonality, the authors argued that:

“The mutual seasonality of influenza and these environmental variables makes it especially important to distinguish causal interactions from spurious correlation.”

“... .. we are concerned here with a more specific problem of distinguishing driving effects from mutual seasonality.”

It is exactly this mutual seasonality and potentially spurious correlation issue that motivated the authors to adopt the CCM approach and seasonal surrogate tests to “control for” seasonality for valid inference of the short-term effects of environmental factors on influenza.

4. The availability of the R-package for CCM makes it very easy to apply this method without real consideration of the topic at hand. I think the authors could substantially improve this paper by focusing on the question, not solely the methodological approach.

It isn't clear to me why the authors chose to do this investigation into ozone, as opposed to another environmental variable. There is much more support for a possible role of precipitation, or even UV radiation, on influenza virus transmission (though a notable lack of population-level studies). Given climate variables are highly correlated, can the CCM model really find a distinct effect of ozone compared to other variables? I would be concerned if, for instance, it found a positive result for any environmental variable. Perhaps the authors can test this.

Re: Using Hong Kong data, we did systematically look into the relation of influenza with an array of environmental factors including all common air pollutants and other factors such as ultraviolet radiation and absolute humidity.

- Ali, S. T., Wu, P., Cauchemez, S., He, D., Fang, V. J., Cowling, B. J., & Tian, L. (05/2018). Ambient ozone and influenza transmissibility in Hong Kong. *The European Respiratory Journal: Official Journal of the European Society for Clinical Respiratory Physiology*, 51(5), 1800369. <https://doi.org/10.1183/13993003.00369-2018>

We found ozone was the only other environmental factor than absolute humidity that can explain some variance of influenza transmissibility in Hong Kong. We found that higher levels of ambient ozone are associated with reduced influenza transmissibility. We have this intention of replicating this finding in different climates and settings, by more rigorous causal inference methods.

5. Why do the authors only analyze data from 28 states? To my knowledge, influenza reporting and surveillance at the state level is very good – shouldn't there be more data? (I just checked and could see flu data from Hawaii and Maine at least). Are states with "missing" values because cases are in fact zero? Also, why do the authors only take data from 2010-2015, when there is data available up until the present. Wouldn't this increase the power of their study?

Re: " Flu_{proxy} ", the proxy measure of influenza activity in the community, is the product of weekly proportion of influenza-like-illness (ILI) consultations among all outpatient visits and the proportion of influenza-positive specimens among all the specimens tested in the lab: $Flu_{proxy} = (Flu_{positive}/Flu_{total}) * (ILI_{outpatient}/Outpatient_{total})$. These four columns of data were consistently available only for 2010-2015 on this website:

<https://www.cdc.gov/flu/weekly/index.htm>.

Yes, your guess was right --- indeed, a lot of missing values in many states were in fact zeros in non-influenza seasons. Restricting our data analysis to influenza season, we now are able to include 46 states for data analysis; the other 4 states (FL, NJ, RI, and VT) had not complete data for even influenza seasons alone in the study period.

6. Can the authors run any other basic statistical tests to demonstrate the ozone link? For

instance, can you show that states with higher ozone have lower transmission. Or that a year with particularly high ozone, has lower transmission. One way to do this would be to use a simple regression model. The authors could calculate transmission for each point of time and location in the dataset, put transmission on the left-hand side and ozone on the right, with controls for time and location.

Re: Convergent cross-mapping (CCM) can easily mislead readers to think that it is a method of spatial analysis. It is not.

“Cross-mapping” here really means cross-prediction over time --- if the reconstructed manifold from Y time series can cross-predict X, then Y must have some ‘imprint’ from X, therefore X is a coupling driver of Y from the dynamical system perspective. The CCM causality test was conducted within each state, and then meta-significance level was estimated using Fisher’s method.

Besides CCM, we now run another two statistical methods to investigate the ozone link. One is PCMCi+, which conducts conditional independence tests under a stochastic probabilistic framework to sketch a causal graph of both contemporaneous and lagged links underlying the targeted system. While state-by-state graphs were generated, a nation-wide graph estimate was obtained by concatenating the state-level time series as a whole before analysis.

Following the reviewer’s suggestion, we also added the traditional regression method of GLM with a quasi-binomial link to estimate the association of O₃ with influenza activity for each state. Robust control for time trend and seasonality was performed. The state-specific regression coefficients were further summarized using random-effect meta-analysis.

Please see Lines 433-550 in the expanded *Methods* section for details. Altogether, the GLM results together with PCMCi+ graph are consistent with our original finding that O₃ is a negative driver affecting influenza activity (please see *Lines 146-178, Table 1 & Figures 4 & 5*).

7. Line 98-99: if we were looking at the effect of climate on transmission, wouldn’t we expect is to be static over time? It seems a weakness of CCM that is can’t capture this? Similarly for Figure , what does it meant that the effect size varies across space? Why would we expect this to vary? It doesn’t seem to vary in a way that corresponds to either influenza or ozone – is this just a failure of the ozone model to capture spatial differences?

Re: It is a strength of CCM by not assuming a static or linear relationship. We expect it to vary because it can be modified by other environmental or socioeconomic factors, over time and space. If we see a clear-cut spatial pattern of the magnitude of ozone effect on influenza, we can infer the effect modification by spatial factors; if we don’t see a spatial pattern of ozone effect size, it does not negate the ozone effect on influenza per se.

8. Line 144; Again, I don’t quite understand why the O3 effect estimates vary over time –

does this mean that if influenza = beta*ozone, this beta is varying, or do you just mean that there is a constant effect of ozone on influenza but ozone varies seasonally? I guess it isn't clear if the beta estimate is varying or if it's just ozone. If I assessed the influence of a completely random variable on influenza, I could make the argument it drove influenza by saying that the effect estimate varied over time and space.

Re: O₃ effect estimates vary over time means the coefficient beta changes overtime. This is due to the different assumptions of methods. In the linear and stochastic relationship of 'influenza = beta*ozone', beta does not vary; there is only one best estimate for beta. This is the case for our GLM method, where beta is a regression coefficient. In our PCMCi+ method in this manuscript, we also assume linearity and use partial correlation as a proxy to infer the magnitude of ozone effect on influenza. In the EDM method, however, we don't assume linearity. The degree of nonlinearity is indicated by the parameter θ , which is estimated by data. In other words, instead of assuming linearity, in EDM we adopted a data-driven approach to decide if the relationship is linear, and if not, to what extent the relationship is non-linear.

“As θ increases, it becomes a more local regression wherein neighbors are identified and predictions increasingly rely on the local information in a reconstructed manifold.” [Ref: Li_2021].

“... as θ increases, predictions become more sensitive to the nonlinear behavior of a system by drawing more heavily on nearby observations to make predictions. In other words, predictions become more state-dependent ...” [Ref: Li_2021].

“A nonlinear system evolves in state-dependent ways, such that its current state influences its trajectory on a manifold M (that is, an unstable process). Conversely, linearity exists if the trajectory on M is invariant with respect to a system's current state.” [Ref: Li_2021].

CCM itself evaluates only causality qualitatively, but lacks the ability to estimate the magnitude of causal effects. The magnitude of the dynamically state-dependent causal effect is inferred by S-map coefficients, which approximate the Jacobian matrix elements (partial derivatives $\partial Flu/\partial O_3$) in the multivariate state space (See the 'S-map' section in manuscript).

- Li, J., Zyphur, M. J., Sugihara, G., & Laub, P. J. (2021). Beyond linearity, stability, and equilibrium: The EDM package for empirical dynamic modeling and convergent cross-mapping in STATA. *The STATA Journal*, 21(1), 220–258. <https://doi.org/10.1177/1536867X211000030>

Minor comments:

9. Line 53-55. I think the authors could better summarize these studies. I'm not sure what "owns virucidal action" means? It seems the evidence linking ozone to influenza is somewhat weak here?

Re: We have rewritten this part to better summarize the mechanistic evidence linking ozone to influenza. Please see the revised text in the manuscript.

In *Introduction* section, *Lines 56-61*:

“Recently, a negative association between ambient ozone (O₃) and influenza transmissibility was also reported in Hong Kong⁸. This human population finding of Hong Kong is consistent with some available laboratory and clinical evidence indicating that the commonly known air pollutant and pulmonary irritant, O₃, not only exhibits virucidal potential through its oxidizing power^{9,10}, but also primes immunity against viral infection¹¹⁻¹³.”

References:

8. Ali, S. T. *et al.* Ambient ozone and influenza transmissibility in Hong Kong. *Eur. Respir. J.* **51**, 1800369 (2018).
9. Murray, B. K. *et al.* Virion disruption by ozone-mediated reactive oxygen species. *J. Virol. Methods* **153**, 74–77 (2008).
10. Tanaka, H., Sakurai, M., Ishii, K. & Matsuzawa, Y. Inactivation of influenza virus by ozone gas. *IHI Engr Rev* **42**, 108–111 (2009).
11. Bocci, V., Borrelli, E., Travagli, V. & Zanardi, I. The ozone paradox: Ozone is a strong oxidant as well as a medical drug. *Med. Res. Rev.* **29**, 646–682 (2009).
12. Michaudel, C. *et al.* Ozone exposure induces respiratory barrier biphasic injury and inflammation controlled by IL-33. *J. Allergy Clin. Immunol.* **142**, 942–958 (2018).
13. Bonilla, W. V. *et al.* The alarmin interleukin-33 drives protective antiviral CD8+ T cell responses. *Science* **335**, 984–989 (2012).

10. Fig 2. I’m not clear on what the “0” line represents here. Improvement compared to what baseline scenario? I think this could be better explained in the text (lines 82 -93).

Re: The original Fig. 2 (now supplementary **Figure S3**) showed results of multivariate forecast improvement (MFI) analysis, which was to see whether the addition of a driver variable, say ozone, can improve the forecasting of influenza. An improvement of zero would nullify the hypothesized driving role. Please see the relevant updates in the Supplementary File 1 (*SF1*).

In *Supplementary File #1 (SF1)*, *Lines 84-97*:

“The concept of MFI test is that: considering two time series X_t and Y_t , if variable X causes variable Y , then better forecast of Y should be obtained by incorporating the information from X and Y simultaneously (i.e., multivariate model) rather than by using the information from Y only (i.e., univariate model). Here, the forecast skill is measured by correlation coefficient (ρ) between the predicted and observed values of Y_t time series. And the improvement in forecast skill is quantified by calculating $\Delta\rho_{MFI} = \rho_{multi} - \rho_{uni}$, that is, the difference between the forecast skills from multivariate and univariate SSR models.

To compare how well the putative drivers alone or in combination could better predict the influenza intensity, we used nonparametric method to examine whether the overall distribution

of $\Delta\rho_{MFI}$ from each state is greater than 0 and also different from each other. One-sample Wilcoxon test and paired two-sample Wilcoxon test were used as appropriate respectively.”

In *SFI*, **Figure S3** legend:

“The forecast improvement (denoted as $\Delta\rho_{MFI}$) indicates the improvement in forecast skill (ρ) by adding embedding coordinates of driving variable(s) in predicting the response variable.”

Reviewer #2 (Remarks to the Author):

1. The authors discuss existing laboratory and clinical evidence for effects of ozone (O3) on influenza infections, motivating their population-level investigation of a causal relationship. While the analysis only spans states within the continental U.S., and thus is only looking at seasonal temperate climates, the authors note previous population-level evidence in Hong Kong. Thus, the study is certainly of sufficient scope and scholarly merit for publication. The analysis are presented in a clear, step-by-step manner and the authors have done due diligence in including code and data to enable close examination and reproduction of the calculations and results.

Re: We appreciate very much your constructive comments here that have been instrumental for us to prepare this revised manuscript.

2. The concerns I have may well be easily resolved, but I do have a major concern about some of the details of Figure 1 that I believe point to an error or inconsistency in a piece of the calculations (which I provide detail on below).

Re: We thank again the reviewer for the extensive input and thoughtful comments that have been discussed and addressed in detail as below.

General questions:

3. Why is the analysis focused on 2010-2015? Is some portion of the data not available earlier?

Re: Yes, we would like to have longer time series beyond 2010-2015. But the state-level influenza data online had not consistent format over the past years; the only available time period of a consistent variable and data format for all states in the USA was 2010-15. Before 2010 or after 2015, the influenza data were not easily accessible for us to estimate the important parameter of “*Flu_{proxy}*”, a proxy measure of influenza activity in the community, which is computed as the product of weekly proportion of influenza-like-illness (ILI) consultations among all outpatient visits and the proportion of influenza-positive specimens among all the specimens tested in the lab: $Flu_{proxy} = (Flu_{positive}/Flu_{total}) * (ILI_{outpatient}/Outpatient_{total})$. These four columns of data were consistently available only for 2010-2015 on this website:

<https://www.cdc.gov/flu/weekly/index.htm>.

In short, before 2010 or after 2015, there were open online data on influenza, but the formality of data forbade us from extracting the necessary 4 variables to estimate Flu_{proxy} .

4. Does O3 particularly help explain the deviations of influenza from seasonality? (My understanding is that O3 has a less pronounced seasonal signature in most or all states than T or AH).

Re: All the 3 environmental factors, including O3, have pronounced seasonality in most states (see **Figure 1**).

Yes, it is O3 that particularly help explain the deviations of influenza from seasonality. This was indicated in CCM seasonal surrogate testing results presented in **Figure 2** and **Table 1**.

Specific questions/concerns:

5. Lines 102-103: the term “unit-free” leaves some ambiguity here. My understanding from other parts of the manuscript is that these are relative to the standard variation of the observed dynamics of those variables, but I think it would be good to state that here for clarity.

Re: Thanks for such careful review. We agree. For clarity, we have modified the “unit-free” statement and replaced “unit” with “standard deviation” in the new manuscript.

In *Results* section, *Lines 138-144*:

“Prior standardization of time series (i.e., subtract the mean and divide by the standard deviation [SD]) was performed to ensure an equal weighting of variables in multivariate SSR. ... When aggregated, the median effect size is -0.107; that is, one SD increment in O₃ (8.3 ppb) causes a -0.107 SD decrease in logit-transformed influenza intensity in the following week.”

6. Lines 142-145: I’m not sure this argument is entirely correct. The inhibiting effect of O3 would be a constant for an individual not the population, no? but shouldn’t be expected at the population level but rather should depend on the current number of cases. If there are 0 cases of influenza, then a change in O3 should have no effect on the cases next week. Indeed, the fact that it shows a highest effect in winter may simply be just because cases are high. Doing an appropriate normalization might be tricky, though, since the cases time series contain 0 counts.

Re: We agree and have removed the statement that ozone effect might be stronger in winter than summer. That statement was very likely wrong because of the zero counts in winters in our earlier analysis. Now, we exclude the non-influenza season data and focus only on the influenza season data so that we are able to include more states for analysis (46 now vs. 28 earlier). In the updated results, we no longer see that artifact of ‘ozone effect might be stronger in winter than summer’. Thanks a lot for the constructive comments.

7. Line 303: An S-map parameter of 0.09 is stated, which would represent fairly weak nonlinearity. However, looking into the SI, it seems this probably supposed to read “0.9” not “0.09”.

Re: Thanks for flagging this. The value 0.09 was a typo; it should be 0.9. In our original manuscript, the nonlinear tuning parameter, θ , for S-map analysis was fixed as 0.9 following the work of Deyle et al. (2016) which studied the global environmental drivers of influenza (see reference below). In this new manuscript, we choose θ in a data-driven manner that maximizes the univariate S-map forecast performance using leave-one-out cross-validation over the whole time series of individual states of USA. Please see details in Supplementary Table S2.

- Deyle, Ethan R., M. Cyrus Maher, Ryan D. Hernandez, Sanjay Basu, and George Sugihara. 2016. “Global Environmental Drivers of Influenza.” *Proceedings of the National Academy of Sciences of the United States of America* 113 (46): 13081–86.

8. Figure 2: It is not clear how the results are being displayed. My best guess is that these are standard box-and-whisker plots with boxes on 25th and 75th quantiles across the samples in each group, and that each sample represents the measured forecast improvement for a single state included in the analysis. The methods relevant to forecast improvement are also ambiguous. Based on the example calculations shown in the SI I am inferring each state was analyzed separately. In both cases, it should be clearly stated in the caption and methods. Additionally, if the analysis was done on a state-by-state basis, it is odd to me that the state-by-state results are shown for CCM and not forecast improvement. I’d request these be included in the supplement.

Re: The significance test for MFI analysis, on a state-by-state basis, has been included in the supplement (supplementary Figures S4). The results of MFI analysis largely coincide with the main-text CCM results, providing further confidence that ambient O₃ is a likely more direct environmental driver of influenza dynamics in the USA.

Figure S4. Multivariate forecast improvement (MFI) on influenza activity by putative environmental drivers (ozone [O₃], absolute humidity [AH], temperature [T]) at 1-week lag in the states of USA. Panel A shows the observed MFI skills (as circles), $\Delta\rho_{MFI}$, and their null distribution tested from 1,000 seasonal surrogates (as line ranges). Panel B shows a summary of state-specific $\Delta\rho_{MFI}$ values in violin plots. Circles are filled to signify the measured $\Delta\rho_{MFI}$ for each state exceeding 95% of its null values. Meta-significance estimate for the nation (P_{meta}) is then tested by summing the logs of state-level P values. The MFI test is deemed significant if $P_{meta} < 1.0 \times 10^{-3}$.

9. Methods. My thanks to the authors for including markdown of core calculations in the supplement. Unfortunately, it did not appear that the `dat_wi.rda`, `analysis_wi.R`, and `functions_wi.R` were actually included in the manuscript files available to me. Nevertheless, the markdown code and `xlsx` files that were available were quite helpful.

Re: We thank the reviewer so much for checking on our reproducibility package. The data and

codes for reproducing our analysis have been uploaded to GitHub:
<https://github.com/PeiZhang0925/USA-FLU>.

10. In reviewing the markdown, one piece of the analysis did stick out, which was how the univariate embedding dimension is determined with out-of-sample predictability, splitting the time series so that the first 2/3 are in-sample training set and the last 1/3 are the out-of-sample test set for predictions. Given that none of the other calculations appear to be done with the same out-of-sample split (including the optimal E calculations for CCM), it seems like an odd choice that will just increase the variance in the results. Determining E on a relatively small prediction set seems like a good way to under embed the system. Unless there is a strong justification I'm missing, I'd strongly recommend that the authors simply use leave-one-out cross-validation for determining E as they do with the remainder of the EDM calculations. This should give a more robust result that should be expected to hold up better e.g. to subsequent testing on newer data.

Re: This is a valuable insight. In our current iteration of analyses, all the predictions including determination of univariate embedding dimension have used the leave-one-out cross-validation strategy. Please see the changes in the revised manuscript.

In *Methods* section, *Lines 383-389*:

“The value of E was chosen over the range of 2 to 6 where the maximum of univariate predictability is achieved via leave-one-out cross-validation (**Table S2**). The lower limit of $E=2$ was specified in order to embed at least one external variable to reconstruct multivariate manifold; the upper limit of $E=6$ was specified because the maximum E should not be larger than the square root of the consecutive time series data length⁶², which was 35 or 34 weekly data points in influenza season in the current study.”

11. Additionally, I was unable to find the exact calculation for obtaining the proxy measure of influence (main text lines 203-213) from the data contained in the supplemental *.xlsx file. That file contains "flu", "flu_total", "ILI", and "ILI_total". My best interpretation of the text is that $fluP = (flu/flu_total) * (ILI/ILI_total)$.

Re: Yes, this is how we computed the variable influenza activity (denoted as “ Flu_{proxy} ”) for analysis. We have included the equation in the *Methods* (please see *Lines 305-306*).

12. This brings me to my biggest question about the main findings. Several details of what is shown in Fig. 1 were a bit surprising to me so I did some digging to try to understand better. In particular, I was very surprised to see in Maryland and Delaware that the seasonal surrogates showed a distribution centered on 0. This surprises me for two reasons. (1) My expectation that influenza and climate in these states is markedly seasonal. Indeed, when I looked at the data in the .xlsx file the linear correlations between “AH” and “flu”, or “AH” and “fluP” (as calculated by my best guess described above) definitely

reflect this for Delaware $\text{cor}(\text{AH}, \text{flu}) = -0.48$; $\text{cor}(\text{AH}, \text{fluP}) = -0.42$. (2) My other expectation is that influenza and climate are strongly correlated to nearby states that show very different surrogate results, namely New York, Pennsylvania, West Virginia, and Virginia. Indeed checking correlations, the climate variables have extremely high correlations ($\text{cor}(\text{AH_Delaware}, \text{AH_NewYork}) = 0.99$), and the influenza high as well ($\text{cor}(\text{fluP_Delaware}, \text{fluP_NewYork}) = 0.80$).

Re: Thanks so much. Please see below our response to Questions 12 and 13 together.

13. Given all this, I have to believe there is some sort of error or inconsistency in the surrogate calculations. My immediate guess is that it is related to missing data in the time series. I notice that Delaware and Maryland are missing 42 and 39 points, respectively, meaning they are amongst the gappiest of the states. For that matter, South Carolina, Montana, Louisiana, and Oregon all also show very low surrogate ρ_{ccm} and are the other states with a large number of missing values. From the markdown it is not clear to me how missing values are handled with the surrogate procedure and the EDM calculations themselves, so that is as far as I can follow my guess. Not having access to state-by-state “forecast improvement” results, it’s hard for me to assess if there might be similar artifacts in that piece of the analysis as well (motivation for my request above to include those at least in the supplement). I would ask the authors check that carefully. That said, it seems like if there are issues it is just with a few of the states, so I would guess that the overall results of the paper won’t significantly change.

Re: We appreciate a lot these extremely insightful and constructive comments, which helped us in the new rounds of data analysis and paper writing. Indeed it was partial missingness in several states that led to the inconsistency in seasonal surrogate time series generation and so unfair significance test in the previous manuscript. In this new manuscript, we analyze only the data of influenza season (that is October through May) in the USA. Finally, out of the 50 states, 46 were available for analysis (FL, NJ, RI, and VT were excluded due to influenza data missingness). Please see our revised *Methods* section for details (*Lines 327-333*).

Reviewer #3 (Remarks to the Author):

1. This paper examines the role of ozone on the transmission of influenza in the continental United States. The authors employ causal framework through EDM modeling of environmental variables and state-level influenza data.

The authors thoroughly explain a complex modelling system and provide evidence of the involvement of ozone in the causal pathway for population-level influenza cases.

Re: We appreciate the reviewer’s positive comments very much.

3. Can the authors provide a causal diagram to illustrate the role of ozone in the causal pathway? Given that temperature influences the amount of ozone and also influenza, is temperature a mediator or a confounder in the relationship?

Re: Thank you for the suggestion. We have now added the the causal network discovery

method PCMCI+ which produced a causal diagram (i.e., **Figure 4** in the revised manuscript). Temperature may influence influenza activity indirectly, mediated through ambient O₃ levels at lag 0.

Figure 4. Graphs estimated by PCMCI+ for individual states (panel A) and USA (panel B) based on the dynamic data of environmental measurements (ozone [O₃], absolute humidity [AH], temperature [T]) and influenza intensity (Flu). Curved and straight edges represent the lagged and contemporaneous causal dependencies, respectively; the number on the curve indicates a lagged relationship in weeks. Node colour denotes autocorrelation strength (i.e., auto-MCI [Momentary Conditional Independence] value); edge colour depicts the causal strength (i.e., cross-MCI) estimated via partial correlation. The hyperparameter significance level (α_{PC}) is set as 0.05 for individual states and 0.001 for the nation-wide analysis.

4. Taking a state-wide average of environmental variables has some limitations - there is certainly a lot of within-state variability that is lost by doing this. Can the authors comment on this limitation and the ability of their method to account for this? Would we

expect data at a finer spatial resolution to give similar results?

Re: We acknowledge this limitation of not being able to address within-state variability in environmental and influenza data. Finer spatial resolution is expected to further corroborate the state-level findings. Using the same methodology here for USA, we have also analyzed data for the single city of Hong Kong and generated similar findings on ozone and influenza. A separate manuscript is being prepared.

Minor edits

5. line 54: “exhibits” might be better than “owns” in this sentence; line 205: remove “should”

Re: Thanks for the careful review. We have modified the text as suggested.

REVIEWER COMMENTS

Reviewer #1 (Remarks to the Author):

I reviewed this manuscript originally a few years ago. The crux of my previous review was that the analysis performed was insufficient to identify ozone as a driver on influenza. Because climate variables are highly correlated, and both diseases and climate cycle seasonally, advanced statistical techniques are required to pinpoint whether a specific climate variable really does drive transmission. It is relatively easy to get strong significant findings of an association if seasonality, trends and other variables are not appropriately controlled for.

In my previous review, I raised concern about convergent cross mapping as a method to detect an environmental signal. There have been several papers on this topic e.g. Baskweville and Cobey 2016,2017. The authors have since updated their manuscript and applied two other methods. I commend the authors for applying additional inference techniques however I remain unconvinced of the results. One of the new methods employed "PCMCI+" is a rarely used approach and appears to have similar issues to convergence cross-mapping. I would like to see the results repeated with a more straightforward approach; the GLM method is a more standard technique, but I still think this model needs additional controls and the current application is insufficient to remove bias.

How are the authors pooling the estimates for the GLM analysis? When I look at the state-level GLM results, the sign of the ozone coefficient switches from positive to negative depending on location, which isn't particularly reassuring. A more standard approach would be to run a panel regression using fixed effects. The authors can implement this e.g. in R using the lfe package, `felm` regression. A panel regression produces a pooled estimate without having to run regressions separately for each state (and then pooling them somehow?).

The authors should run a fixed effect panel regression model where outcome variable is logged cases and dependent variables are the three lags of humidity, temperature and ozone as shown in Table 1. This regression should include the whole dataset across states but include a fixed effect at the state level and fixed effect for year. These fixed effects account for mean differences in e.g. reporting across states as well reporting issues that may trend over time. The authors should then repeat this exercise including a seasonal dummy e.g. weekly indicator variable which accounts for any other factors (school semesters/holidays) that may drive transmission on a seasonal scale. If both of these regressions are significant and with the same sign on ozone, then the result would be more convincing.

The author's GLM results find a role of temperature, absolute humidity as well as ozone. How important is the ozone effect compared to these other two drivers?

In my previous review I raised this question: "The authors argument for the O3 driver rests only on the CCM results, but is there other evidence? For instance, we know that influenza dynamics closer to the tropics e.g. Hawaii, tend to be more persistent. Is this the same with ozone seasonality in the tropics? Do spatial patterns of ozone match spatial patterns of influenza?" The authors have not really answered this and instead copy-pasted quotes from a different manuscript. I would like to understand whether spatial patterns of ozone can explain spatial patterns of influenza - this is not unreasonable if ozone is an important driver. Or is it not really that important compared to AH and temp?

Reviewer #2 (Remarks to the Author):

I appreciate the authors thoughtful responses to my initial comments and see that they have taken considerable additional work to address other reviewer comments. By my reading this is a robust analysis with multiple angles of support and reasoned arguments connecting the pieces. I have gone through the new code as best as time allows and just have one minor concern about a methodological change the authors describe in focusing exclusively on segments of the year falling between October and May:

“Now, we exclude the non-influenza season data and focus only on the influenza season data so that we are able to include more states for analysis”

In my experience, data gaps and discontinuities need to be treated with utmost care when applying the current incarnations of the rEDM package. Looking through the code, it looks like a function `make_pred_nozeroL` is mediating that behavior by constructing appropriate library segments so that the EDM calculations do not depend on any data outside of October – May. As far as I can tell this is a reasonable and correct approach, but as I said it can be a tricky business that needs the utmost care. I would suggest the authors just make one additional test that it is working as intended, then, by modifying data just before and just after the winter period which should be ignored in the calculations as constructed and verify that the results do not change. So long as the authors can check this point to their satisfaction, I do not ask to see a revision or resubmission.

Response to reviewers' comments --- NCOMMS-21-05266B

Kindly note that the texts in **black** are the comments from the reviewers, and our responses are positioned right after the comments and colored in **blue**. The changes made in the revised manuscript following the reviewers' comments and suggestions are also **highlighted**.

Reviewer #1 (Remarks to the Author):

I reviewed this manuscript originally a few years ago. The crux of my previous review was that the analysis performed was insufficient to identify ozone as a driver on influenza. Because climate variables are highly correlated, and both diseases and climate cycle seasonally, advanced statistical techniques are required to pinpoint whether a specific climate variable really does drive transmission. It is relatively easy to get strong significant findings of an association if seasonality, trends and other variables are not appropriately controlled for.

Re: Thanks for your rigorous evaluation of our research work and manuscript. Indeed, we share the same concern as the reviewer during the process of scientific inquiry.

Thanks to the constructive comments from you, other reviewers, and the editor, we are propelled to adopt 3 methods with disparate theoretical assumptions and hidden biases for data analysis which complement each other and enhance the coherence of evidence. While scrutinizing the relationship between O₃ and influenza by comprehensive data analyses, we expanded our discussion of the literature to include the latest mechanistic evidence from the realm of immunology, thereby providing a more thorough interpretation of the results, as well as highlighting the novelty and significance of our findings in the broader scientific context.

In my previous review, I raised concern about convergent cross mapping as a method to detect an environmental signal. There have been several papers on this topic e.g. Baskweville and Cobey 2016,2017. The authors have since updated their manuscript and applied two other methods. I commend the authors for applying additional inference techniques however I remain unconvinced of the results. One of the new methods employed "PCMCI+" is a rarely used approach and appears to have similar issues to convergence cross-mapping. I would like to see the results repeated with a more straightforward approach; the GLM method is a more standard technique, but I still think this model needs additional controls and the current application is insufficient to remove bias.

How are the authors pooling the estimates for the GLM analysis? When I look at the state-level GLM results, the sign of the ozone coefficient switches from positive to negative depending on location, which isn't particularly reassuring. A more standard approach would be to run a panel regression using fixed effects. The authors can implement this e.g. in R using the lfe package, felm regression. A panel regression produces a pooled estimate without having to run regressions separately for each state (and then pooling them somehow?).

The authors should run a fixed effect panel regression model where outcome variable is logged

cases and dependent variables are the three lags of humidity, temperature and ozone as shown in Table 1. This regression should include the whole dataset across states but include a fixed effect at the state level and fixed effect for year. These fixed effects account for mean differences in e.g. reporting across states as well reporting issues that may trend over time. The authors should then repeat this exercise including a seasonal dummy e.g. weekly indicator variable which accounts for any other factors (school semesters/holidays) that may drive transmission on a seasonal scale. If both of these regressions are significant and with the same sign on ozone, then the result would be more convincing.

Re: Developed in 2020, PCMCI+ is part of the constraint-based causal discovery methods family, employing conditional independence tests to infer causality ^[1,2]. It improves upon its predecessor, PCMCI, by also identifying contemporaneous links. To date, PCMCI has seen broad application across fields (with a total of 618 citations) ^[3-5]. PCMCI+ might be relatively less utilized as an advanced version of PCMCI developed 3 years after, but is attracting growing interest, evidenced by its steadily increasing annual citations. Both simulation experiments and real-world case studies have demonstrated PCMCI+'s effectiveness. We anticipate that its use will continue to expand in the years ahead.

Generalised liner model (GLM) was a classic approach used in the realm of environmental epidemiology to test environment-health relationships with observational dynamic data. In our paper, GLM was applied to the state-level data while adjusting for temporal trend and seasonality; then, the state-level coefficients were pooled by a meta-analysis model.

Here, as suggested by the reviewer, panel regression using fixed effects (realized via `lfe::felm()` function in R) is performed to test the robustness of findings generated by GLM analysis. During FELM regression, fixed-effect variables for “state” and “year” have been included in the model to account for state-specific variables and time-varying unobserved factors that may influence the baseline influenza intensity, respectively. After further adjustment of seasonality and holiday factors (as dummy variables), we detected consistently significant relationship between O₃ and influenza in a negative manner. Please see the details below extracted from our revised manuscript.

In Supplementary File 1, Lines 137-168:

“In the main manuscript, we used classic time series regression method, generalised linear model (GLM) with a quasi-Binomial link (i.e., logit function) to estimate the statistical association of O₃, AH, and T with influenza intensity at each state, and then pooled the state-level coefficients via meta-analysis. Here, as a robustness check, we used another regression-based method, FELM, by including “state” as fixed effects during nation-wide analysis, to control for time-invariant factors specific to each state (e.g., socioeconomic or policy differences) that may influence the baseline influenza intensity ^[9]. Calendar year was included as fixed effects in the model to control for time-varying unobserved factors (e.g., potential reporting issues that may trend over time). Then, the core FELM was re-run by further controlling for seasonality and school semesters/holidays (as dummy variables) that may bias

the link of interest on a seasonal scale. To account for strong autocorrelation caused by disease transmission, we took the logarithm of 1-week and 2-week lagged outcome variables (i.e., $\log(Y_{t-1})$ and $\log(Y_{t-2})$) as covariates in the model. When estimating the relationship of O₃ with influenza, the same-week AH and T are simultaneously included in the model as a linear term to control for potential confounding. The “lfe” (version 2.9.0)^[10] was adopted to fit the fixed-effects linear regression model.

Table S3 summarizes the lag-specific statistical relationships between environmental factors and influenza activity from the FELM regressions, by contrast with those from GLM regressions. Ambient O₃ was found to reduce influenza activity ($P < 1.0 \times 10^{-3}$) at lag 1 (week), consistently by the two regression-based methods. While GLM revealed a negative link between AH and influenza at lag 1, FELM detected none. The negative association of 2-week lagged air T with influenza, though presented simultaneously by FELM and GLM regressions, is not supported by CCM and PCMCI+ analyses shown in the main manuscript (**Table 1**). **Figure S4** further visualizes the state-wise regression results by FELM at the lag 1. Panel A showed that the 1-week lagged statistical associations of O₃ with influenza intensity are generally negative at the state level. By conducting nationwide analysis with state included as fixed effects (panel B), one SD increment in O₃ concentration is associated with a reduction of 0.106 (CI: -0.163, -0.048; $P < 1.4 \times 10^{-9}$) in logit-transformed influenza intensity one week after.”

Table S3. Effects of environmental factors on influenza activity estimated by two regression-based methods, based on weekly state-level data of the USA during 2010-2015

	GLM		FELM	
	Effect	P-value	Effect	P-value
Ozone				
Lag 0	-0.016	5.4×10^{-1}	-0.113	6.4×10^{-10}
Lag 1	-0.102	5.9×10^{-5}	-0.106	1.4×10^{-9}
Lag 2	-0.017	3.9×10^{-1}	-0.087	4.3×10^{-7}
Absolute humidity				
Lag 0	-0.037	4.3×10^{-1}	-0.083	8.5×10^{-3}
Lag 1	-0.310	6.7×10^{-8}	-0.076	1.3×10^{-2}
Lag 2	-0.020	6.6×10^{-1}	-0.018	5.6×10^{-1}
Temperature				
Lag 0	0.075	2.6×10^{-2}	-0.195	1.5×10^{-8}
Lag 1	0.076	1.1×10^{-1}	-0.256	2.8×10^{-14}
Lag 2	-0.158	2.0×10^{-6}	-0.257	1.7×10^{-14}

Abbreviations: GLM, generalised linear model; FELM, fixed effects linear model.

Note: In the 2 sets of results, bold values suggest statistically significant relationships with statistical significance of $P < 1.0 \times 10^{-3}$. With GLM, regressions were first performed at state-level

and then pooled by meta-analysis; while with FELM, state-level data were pooled to run nation-wide regression but with fixed effects defined by "state".

**Figure S4. Effects of environmental factors on influenza intensity, at 1-week lag, estimated by Fixed Effects Linear Model (FELM).** Panel **A** shows the state-level point estimates (as circles) of regression coefficients (β) and their 95% confidential intervals (CIs) (as line ranges). Panel **B** shows the nation-wide risk estimates(β with 99.9% CIs) by including "state" fixed effects. Filled circles indicate that the risk estimate is significantly better than the null expectation ($P < 0.05$ and $P < 0.001$ for state-wise and nation-wide analysis, respectively). $O_3$: ozone; AH: absolute humidity; T: temperature.

## Reference

1. Runge, Jakob, Andreas Gerhardus, Gherardo Varando, Veronika Eyring, and Gustau Camps-Valls. 2023. "Causal Inference for Time Series." *Nature Reviews Earth & Environment* 4 (7): 487–505.
2. Runge, Jakob. 2020. "Discovering Contemporaneous and Lagged Causal Relations in Autocorrelated Nonlinear Time Series Datasets." In *Proceedings of the 36th Conference on Uncertainty in Artificial Intelligence (UAI)*, edited by Jonas Peters and David Sontag, 124:1388–97.
3. Krich, Christopher, Jakob Runge, Diego G. Miralles, Mirco Migliavacca, Oscar Perez-Priego, Tarek El-Madany, Arnaud Carrara, and Miguel D. Mahecha. 2020. "Estimating Causal Networks in Biosphere–atmosphere Interaction with the PCMCI Approach." *Biogeosciences* 17 (4): 1033–61.
4. Qu, Yuquan, Carsten Montzka, and Harry Vereecken. "Causation discovery of weather and vegetation condition on global wildfire using the PCMCI Approach." *2021 IEEE International Geoscience and Remote Sensing Symposium IGARSS*. IEEE, 2021.
5. Maisonnave, Mariano, Fernando Delbianco, Fernando Tohme, Evangelos Milios, and Ana G. Maguitman. 2022. "Causal Graph Extraction from News: A Comparative Study of Time-Series Causality Learning Techniques." *PeerJ. Computer Science* 8 (August): e1066.

The author’s GLM results find a role of temperature, absolute humidity as well as ozone. How important is the ozone effect compared to these other two drivers?

Re: O₃ appears to be more important than the other two drivers because it was only O₃ that was consistently found to be a driver in 3 methods of CCM, PCMCI+, and GLM—as well as in FELM (see **Table 1** and **Table S3**). These results on O₃, however, do not necessarily contradict the earlier findings on temperature and humidity as environmental drivers of influenza because it is likely that O₃ is a more direct environmental driver of influenza while the other two variables of temperature and humidity operate indirectly via O₃ on influenza. Indeed, as seen in the graph of **Figure 4B**, temperature affects influenza intensity in a negative manner indirectly through O₃ at lag 0; humidity has a direct negative effect on influenza at lag 1 but it has also indirect but positive effect through O₃.

In my previous review I raised this question: “The authors argument for the O₃ driver rests only on the CCM results, but is there other evidence? For instance, we know that influenza dynamics closer to the tropics e.g. Hawaii, tend to be more persistent. Is this the same with ozone seasonality in the tropics? Do spatial patterns of ozone match spatial patterns of influenza?” The authors have not really answered this and instead copy-pasted quotes from a different manuscript. I would like to understand whether spatial patterns of ozone can explain spatial patterns of influenza - this is not unreasonable if ozone is an important driver. Or is it not really that important compared to AH and temp?

Re: It is possible that spatial patterns of O₃ can explain spatial patterns of influenza in the USA,

but this question is beyond the scope of this current manuscript which focus on the acute effects of environmental factors on influenza within two weeks—using the weekly time series data of individual states in the USA. One advantage of time series analysis is that it is not vulnerable to those common confounders in spatial data analysis such as social economic status, demographics, smoking prevalence which do not vary week by week.

Given the note in earlier comment “(Notably, Deyle 2016 actually showed some of this spatial evidence first, before applying CCM)”, we simply wanted to clear the air in our responses that Deyle et al (2016) paper did not examine the spatial correspondence between environmental variables and influenza, which is subject to spurious correlation. Rather, they checked on the correspondence between seasonality of environment and seasonality of influenza (just depicted via map)—they found Spearman correlation between the two was high—which has propelled them to adopt the CCM approach and seasonal surrogate tests to distinguish driving effects from mutual seasonality in the time series analysis.

We do not see a particular spatial pattern of the O₃ effect on influenza estimated at individual state level. As seen in **Figure 3** on the state-specific effect size estimates for O₃ affecting influenza activity at 1-week lag, all states see negative effect size except for the two states of Mississippi and Hawaii though they do not pass the CCM causality tests (**Figure 2A** on MS and HI). We have no speculation on the Mississippi finding, but the Hawaii finding could be confounded by tourism activities that co-vary with both ambient O₃ (more affected by local activities in comparison with other states affected by regional O₃) and influenza activity. Please see our manuscript for clearer understanding.

In Introduction section, Lines 62-66:

“In this work, we used the publicly available weekly state-level data in the USA during 2010–2015 to examine the acute effect of ambient O₃ on influenza dynamics—whether a change in weekly ambient O₃ leads to a change in influenza activity within two weeks in the community when keeping all other variables the same.”

In Results section, Lines 135-136:

“All states see negative effect size except for the two states of Mississippi and Hawaii though they do not pass the CCM causality tests (**Figure 2A**).”

In Discussion section, Lines 259-264:

“It remains an important topic for future studies to decode the nuanced relationships of environmental variables with influenza activity on finer spatial and temporal scale, since factors such as demographic features, social connectivity, **tourism activities (e.g., Hawaii)**, as well as public health interventions can lead to fundamentally different base transmission potential, which may interact with environmental factors to shape the complex influenza

dynamics.”

Reviewer #2 (Remarks to the Author):

I appreciate the authors thoughtful responses to my initial comments and see that they have taken considerable additional work to address other reviewer comments. By my reading this is a robust analysis with multiple angles of support and reasoned arguments connecting the pieces. I have gone through the new code as best as time allows and just have one minor concern about a methodological change the authors describe in focusing exclusively on segments of the year falling between October and May:

“Now, we exclude the non-influenza season data and focus only on the influenza season data so that we are able to include more states for analysis”.

In my experience, data gaps and discontinuities need to be treated with utmost care when applying the current incarnations of the rEDM package. Looking through the code, it looks like a function `make_pred_nozeroL` is mediating that behavior by constructing appropriate library segments so that the EDM calculations do not depend on any data outside of October – May. As far as I can tell this is a reasonable and correct approach, but as I said it can be a tricky business that needs the utmost care. I would suggest the authors just make one additional test that it is working as intended, then, by modifying data just before and just after the winter period which should be ignored in the calculations as constructed and verify that the results do not change. So long as the authors can check this point to their satisfaction, I do not ask to see a revision or resubmission.

Re: We appreciate so much your constructive comments throughout the review process, which have been instrumental for us to refine the analyses and improve the overall quality of this manuscript.

Yes, we are assured that the function of “`make_pred_nozeroL`” works as intended in handling data gaps and discontinuities. Fed with data of flu season (from October throughout May) rather than the whole year data that included zero values, this “`make_pred_nozeroL`” function generates a matrix showing the row indices for each "October throughout May" segment. This matrix is then used to specify the sections of time series to create the libraries.

REVIEWERS' COMMENTS

Reviewer #1 (Remarks to the Author):

The authors have addressed my previous comments. I have no further questions.